# Evolutionary innovation through transcription factor rewiring in microbes is shaped by levels of transcription factor activity, expression, and existing connectivity

**Matthew J. Shepherd[1,2], Aidan P. Pierce[1], Tiffany B. Taylor[1] ***

1 Milner Centre for Evolution, Department of Life Sciences, University of Bath, Bath, United Kingdom,
2 Division of Evolution and Genomic Sciences, School of Biological Sciences, University of Manchester, Manchester, United Kingdom

* t.b.taylor@bath.ac.uk

**Data Availability Statement:** All raw data for this study is available on the Open Science Framework (OSF), and can be accessed at

## Abstract

The survival of a population during environmental shifts depends on whether the rate of phenotypic adaptation keeps up with the rate of changing conditions. A common way to achieve this is via change to gene regulatory network (GRN) connections—known as rewiring—that facilitate novel interactions and innovation of transcription factors. To understand the success of rapidly adapting organisms, we therefore need to determine the rules that create and constrain opportunities for GRN rewiring. Here, using an experimental microbial model system with the soil bacterium *Pseudomonas fluorescens*, we reveal a hierarchy among transcription factors that are rewired to rescue lost function, with alternative rewiring pathways only unmasked after the preferred pathway is eliminated. We identify 3 key properties—high activation, high expression, and preexisting low-level affinity for novel target genes—that facilitate transcription factor innovation. Ease of acquiring these properties is constrained by preexisting GRN architecture, which was overcome in our experimental system by both targeted and global network alterations. This work reveals the key properties that determine transcription factor evolvability, and as such, the evolution of GRNs.

## Introduction

During conditions of environmental upheaval and niche transition events, rapid phenotypic adaptation is essential for evolutionary success [1–3]. To understand patterns of diversification in novel environments, we need to understand why some evolutionary transitions occur more rapidly than others, and what allows some organisms to succeed where others fail. Key to this is understanding the evolution of gene regulatory networks (GRNs), control circuits common throughout the domains of life that determine the magnitude and timing of gene expression [4] in response to environmental and internal signals [5,6]. GRNs are frequent sites of adaptive mutation driving phenotypic evolution [7–9] and often underscore adaptation to—and survival in—new and changeable environments [10–13]. Alterations to GRNs can also enhance

https://osf.io/pcdhx/. Data is also available in supplementary data files where cited. RNAseq files are available on NCBI GEO accession GSE228016.

**Funding:** This work was funded by a Royal Society Research Fellows Grant (RG160491; awarded to TBT) supporting MJS; Royal Society Research Fellows Enhancement Award (RGF\EA\201057; awarded to TBT) supporting APP; a Royal Society Dorothy Hodgkin Research Fellowship (DH150169; awarded to and supporting TBT). The funders had no role in study design, data collection and analysis, decision to publish, or preparation of the manuscript.

**Competing interests:** The authors have declared that no competing interests exist.

**Abbreviations:** AUGC, area under the growth curve; GRN, gene regulatory network; HTH, helix-turn-helix; LB, lysogeny broth; ONPG, o-nitrophenyl-β-D-galactopyraniside; rAUGC, relative area under the growth curve; RpoN-EBP, RpoN-dependent enhancer binding protein; SNP, single nucleotide polymorphism; SOE, strand overlap extension; stRBS, strong ribosome binding site.

drug resistance and stress responses in the face of environmental challenges [14–17]. Determining how these networks evolve and identifying rules governing when and how a regulatory circuit adapts will allow us to better understand their role in determining the evolutionary success of an organism.

Within GRNs, a key mechanism of innovation is transcription factor rewiring, in which transcription factors can gain or lose regulatory connections to target genes creating new network architectures and opportunity for phenotypic innovation [18]. Rewiring events can have dramatic effects on the transcriptome [19] that can drive phenotypic diversification. However, the majority of past studies on transcription factor rewiring involve retrospective experimental dissection of networks that have already diverged [20–25], which allows inference of past rewiring events but does not address the evolutionary factors driving the rewiring process. For rewiring to occur, a transcription factor must first have the potential interaction with non-cognate regulatory targets, allowing regulation of a new gene [26], as is the case for innovation in other proteins [27,28]. The potential for non-cognate interactions comes from built-in homology between paralogous components of different regulatory networks—a consequence of gene duplication and divergence [29]. Non-cognate interactions can become meaningful and drive adaptation if favoured by natural selection [30,31]. However, non-cognate interactions cannot be commonplace within GRNs as this will likely result in dysregulation of gene expression and fitness costs for an organism in the environment to which it is adapted [32,33].

While there exists some empirical evidence of the importance of transcription factor rewiring in driving evolutionary innovation [25,34] and generation of novel transcription factors through gene duplication [35], in each case, it is unknown why the transcription factor in question rewired as opposed to any of the other regulators within their protein family. For example, structurally and functionally similar phage repressor proteins λ cI and P22 c2 differ in their ability to regulate expression from non-cognate sites with λ cI being more evolvable, a property that correlates with higher robustness to mutation [36]. What role—if any—GRN structure plays in these processes and whether the rewired transcription factors were unique in their ability to rewire within each study system is also undetermined. To address these questions, we set out to investigate the evolution of transcription factor rewiring using rescue of flagellar motility via rewiring of NtrC documented by Taylor and colleagues [37]. In this model system, *Pseudomonas fluorescens* SBW25 is engineered to be non-motile via deletion of the master regulator for flagellar synthesis (*fleQ*) and abolishment of biosurfactant production. Placing these mutants in soft agar plates results in strong selection for rescue of motility—bacteria will grow, exhaust available nutrients, and starve, unless they acquire a mutation that restores motility, allowing them access to uncolonised areas of the agar plate [37,38]. Under strong selection for motility, the bacteria rapidly and reliably evolve new regulatory network wiring to rescue flagellar motility (a schematic of these experiments is laid out in Fig 1A) [37] with the same transcription factor (*ntrC*) repurposed to rescue flagellar-driven motility each time, to the exclusion of all other homologous transcription factors within the protein family [39]. However, we do not know the factors that determine this evolutionary preference. To test this, we constructed a double *fleQ ntrC* knockout and placed this mutant under strong selection for swimming motility. This forces evolutionary utilisation of an alternative transcription factor in order to rescue motility and is an established method for unveiling hidden evolutionary pathways [40]. We reveal a hierarchy among transcription factors that are rewired to rescue lost flagellar function, with alternative rewiring pathways only revealed after the preferred co-opted transcription factor, NtrC, is eliminated. Identification of an additional transcription factor capable of rewiring within the same GRN background allows us to investigate rules governing when and where a transcription factor rewires within its GRN and is important for

**Fig 1. Rewiring of RpoN-EBP transcription factors to rescue flagellar motility. (A)** Flagellar motility rescue experiment route map outlines typical progression of motility evolution. Pathway diagrams display key components of the primary (**B,** Taylor and colleagues [37]) and alternative (**C,** this study) rewiring pathways for rescue of flagellar gene expression. Genes are coloured in white, protein components in green. Mutational targets and their effects are shown in red, and whether a mutation occurs in the first or second evolutionary step indicated. Figure created with BioRender.com.

understanding how these regulatory systems innovate during adaptation to environmental challenges.

## Results

### Evolutionary rescue of flagellar motility can occur in the absence of *ntrC* through de novo mutation to an alternative two-component system

When previously challenged to rescue flagellar motility in the absence of FleQ, rewiring almost exclusively occurred through the NtrC transcription factor [37,38]. Perhaps we only see NtrC co-opted to rescue FleQ function because it is the only transcription factor capable of doing this? It is known that within families of homologous transcription factors, there is variation in the ability to bind non-cognate sites [36], so it is possible that NtrC is unique in its ability to rewire. FleQ and NtrC are part of a family of 22 structurally related transcription factors called <u>RpoN</u>-dependent <u>enhancer binding proteins</u> (RpoN-EBPs), many of which are predicted to be more structurally similar to FleQ than NtrC through 3D structural modelling [39]. To identify

if any other RpoN-EBPs were capable of rewiring, the double knockout non-motile *P. fluorescens* (*ΔfleQΔntrC*) was challenged to rescue flagellar motility in 0.25% agar lysogeny broth (LB) plates. Motile zones reemerged in a two-step manner (a slow-swimming variant, then a faster-swimming variant) as typical of previous studies [37]. Motile isolates were sampled and whole genome resequenced at each step. Motility-granting mutations were identified in the gene *PFLU1131* for all first-step motile isolates (*n* = 15), and in 13/15 cases, this was the only mutant gene (S1 File). *PFLU1131* is unstudied outside of our own study system [40,41] and encodes a putative sensor kinase, a protein which—in response to a signal—will modulate the activity of a cognate transcription factor through phosphorylation or dephosphorylation. The cognate transcription factor is typically grouped in the same operon as the kinase, and the PFLU1131 gene is situated in an operon between 2 other genes, *PFLU1130* and *PFLU1132*, which encode a hypothetical GNAT-acetyltransferase and a putative RpoN-dependent transcriptional regulator, respectively. PFLU1132 is a known FleQ-homolog [39] and together with PFLU1131 forms a putative two-component system (a kinase and regulator pair, a common regulatory system in bacteria [42]). The most frequent mutation in *PFLU1131* was an identical in-frame 15-bp deletion (Fig 2A) in the histidine-kinase phospho-acceptor domain (73%). This mutation results in loss of 5 amino acids (368-GEVAM-372) in the protein product (henceforth referred to as PFLU1131-del15). Other mutations included a highly similar 15-bp deletion resulting in loss of 5 amino acids a few amino acids downstream (369-EVAMG-373), as well as a single nucleotide polymorphism (SNP) resulting in A375V. All of these mutations (86% of the first-step mutations) cluster to the same 26 bp of the 1,770-bp *PFLU1131* ORF and result in amino acid changes in a site directly adjacent to the catalytically active H-box [43] (amino acids 376–382; Fig 2A), suggesting a significant effect on the catalytic function of the putative kinase. We additionally constructed a *PFLU1132* knockout in the *ΔfleQΔntrC* background to identify any further RpoN-EBPs capable of rewiring. *ΔfleQΔntrCΔPFLU1132* failed to rescue flagellar motility within the 6-week assay cutoff (S2 File; *n* = 192), suggesting evolutionary rewiring of other transcription factors may not be easily achieved under the conditions we tested.

Together, these results suggest that in the absence of FleQ and NtrC, flagellar motility can be rescued via rewiring of an alternative transcription factor in the same protein family, PFLU1132. To confirm that observed motility phenotypes were dependent on the PFLU1131/2 two-component system, the *PFLU1132* gene was deleted and complemented with and without the presence of the most common first-step kinase mutation (*PFLU1131*-del15). Knockout of *PFLU1132* abolished flagellar motility, and complementation restored motility only in the presence of the kinase mutation (Fig 2B). This experiment was repeated for *ntrBC* and produced the same result (S1 Fig). The *PFLU1131*-del15 mutation also grants flagellar motility when *ntrC* is present (i.e., in a *ΔfleQ* background), so does not depend on the absence of *ntrC* to function (S2 Fig)—ruling out the possibility that the presence of NtrC suppresses the alternative pathway. Transcriptomic analysis of the *PFLU1131*-del15 mutant by RNA sequencing indicates an alteration to the activity of the PFLU1131/2 two-component system that results in a net up-regulatory effect on the transcriptome (Fig 2C). These regulatory changes include up-regulation of the flagellar genes (S3 Fig) and the *PFLU1130/1/2* operon. In sum, in the "primary rewiring route," repeatable mutations in the NtrBC (Fig 1B) two-component system were found to rescue flagellar motility. In the absence of both FleQ and NtrC, we identified the "alternative rewiring route," via PFLU1131/2 (Fig 1C), which was also capable of rescuing flagellar motility.

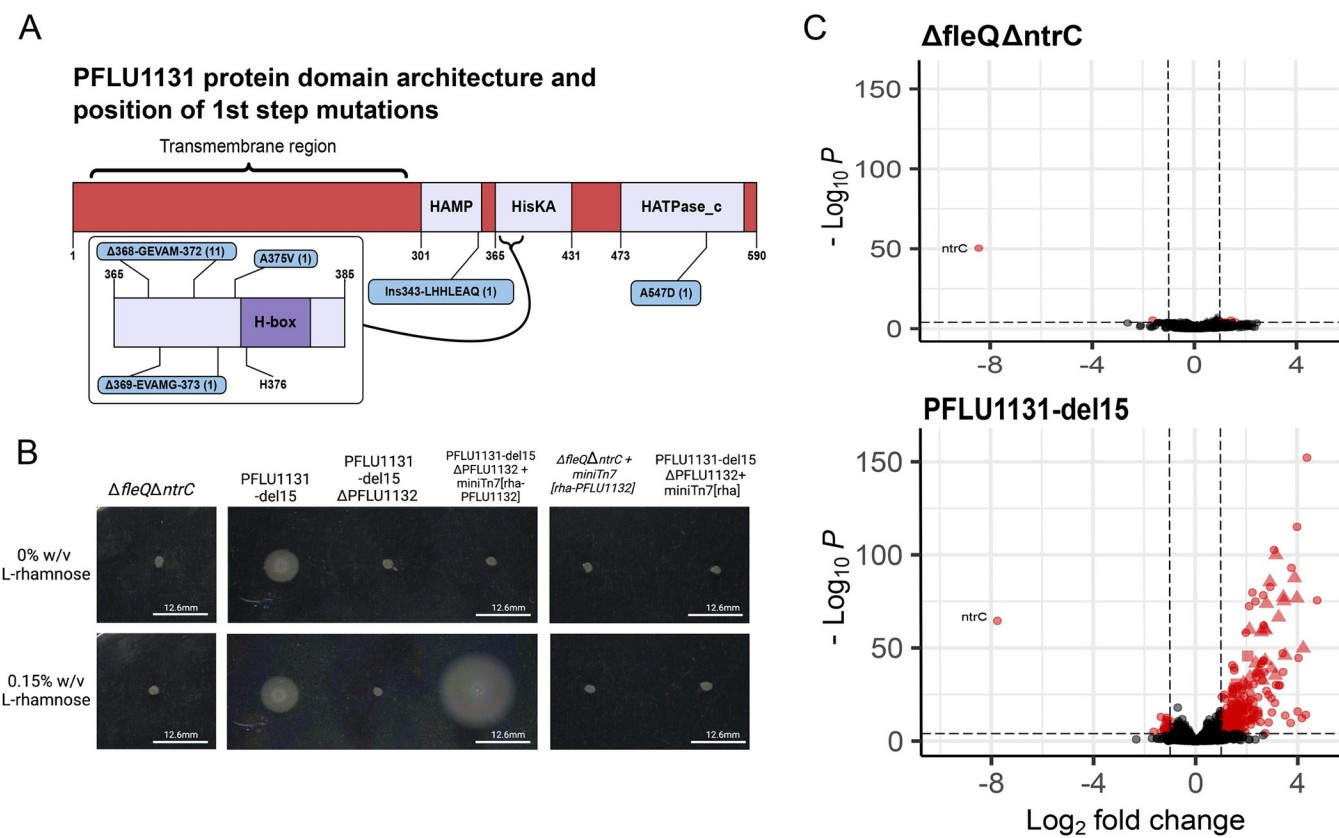

**Fig 2. In a *ΔfleQΔntrC* background, motility rescue initially occurs through of mutations in *PFLU1131*, which have a net up-regulatory effect on the transcriptome. (A)** Diagram of PFLU1131 protein, indicated positions of first-step motility-rescuing mutations. Predicted protein domains (pfam) are indicated in lilac; amino acid positions are indicated as numbers below each domain. Mutations are indicated in blue boxes, with the number of replicate lines gaining each mutation indicated in brackets. **(B)** FleQ-homolog encoding gene *PFLU1132* is essential for rescued flagellar motility in the *PFLU1131*-del15 mutant strain and depends on the presence of the kinase mutation. Scale bars (white) = 12.6 mm. Transcription factor gene *PFLU1132* is deleted and reintroduced as a single-copy chromosomal insertion expressed from an L-rhamnose–inducible promoter system (*rhaSR-PrhaBAD*). The same complementation lacking the *PFLU1131*-del15 mutation as well as an empty expression system transposon were included as further controls. Photographs of motility after 1-day incubation in 0.25% agar LB plates supplemented with or without 0.15% L-rhamnose for induction of transcription factor expression. In all cases, the genetic background for these mutants and constructs is a *ΔfleQΔntrC* background. **(C)** Volcano plots indicating impact of *PFLU1131*-del15 mutation on the transcriptome relative to the *ΔfleQ* ancestor. Red points indicate significantly differentially expressed genes. Triangles indicate flagellar genes; squares indicate *PFLU1131/2* and adjacent genes; and circles indicate all other genes. Vertical and horizontal dashed lines indicate significance cutoff values on both the *x* and *y* axes. Data underlying this figure can be found in S6 File.

## First-step mutations in the alternative rewiring pathway rewire another FleQ homologous transcription factor

In the primary rewiring pathway, NtrC (a homolog of FleQ) was recruited to recover lost FleQ function. This occurred as a two-step process: an initial motility-granting mutation to the gene *ntrB* encoding NtrC's cognate kinase and a secondary motility-enhancing mutation to the helix-turn-helix (HTH) DNA binding domain of the NtrC transcription factor. Similarly, in the *ΔfleQΔntrC* background strain, first-step mutations restored slow motility, showing that alternative rewiring routes were possible, but not utilised in the presence of *ntrC*. In 13/15 first-step mutants, only a single mutation to *PFLU1131* was identified, suggesting that only this mutation is required for flagellar motility to be rescued. To confirm this, the *PFLU1131*-del15 mutation was introduced into the ancestor (*ΔfleQΔntrC*). The resulting engineered strain was motile (S2 Fig). As no mutational change to the DNA binding domain of PFLU1132

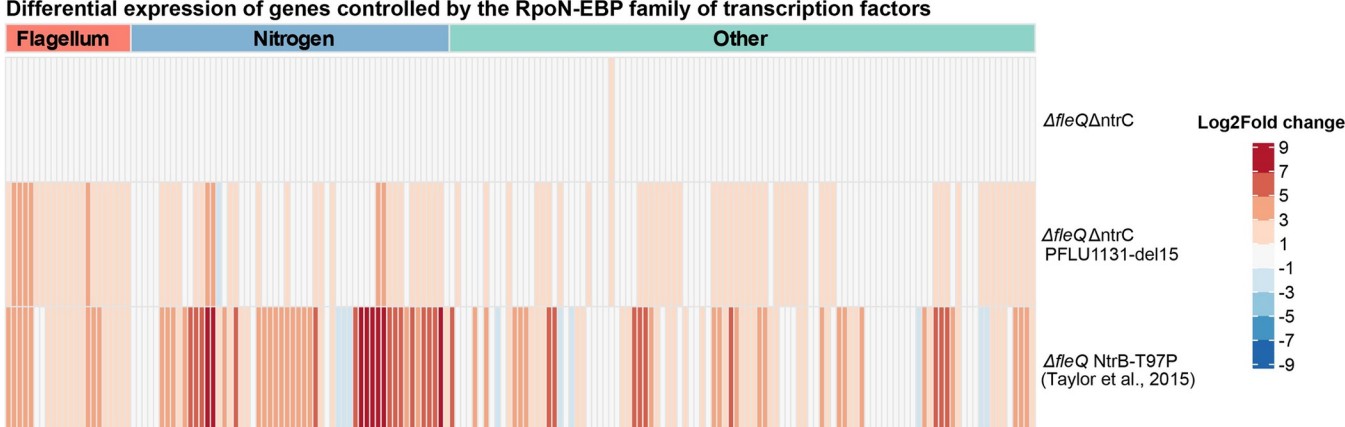

**Fig 3. First-step mutations result in up-regulation of flagellar and other RpoN-EBP controlled genes.** Heatmap of Log2Fold changes in gene expression for RpoN-EBP controlled genes after first-step motility mutations, relative to Δ*fleQ* ancestor. Δ*fleQ*Δ*ntrC* background and *PFLU1131*-del15 mutant (a first-step mutant, also Δ*fleQ*Δ*ntrC* background) differentially expressed genes were determined by RNA sequencing (this work). Δ*fleQ* first-step (NtrB-T97P) differentially expressed genes were identified by RNA microarray and originally reported by Taylor and colleagues [37]. List of RpoN-EBP controlled genes was obtained from Jones and colleagues [44]. Groups of genes by function are indicated by the coloured bars above the plot: red: flagellum; blue: nitrogen; teal: other/unknown. Data underlying this figure can be found in S7 File.

is needed, this suggests a non-specific mechanism through which this transcription factor induces the flagellar genes.

To investigate this, our transcriptomic data were used to assess the impact of the *PFLU1131*-del15 mutation on the expression of a list of genes [44] controlled by all 22 homologous transcription factors (part of the RpoN-EBP family) present in *P. fluorescens* SBW25 [45]. Regulatory changes predictably include up-regulation of the flagellar genes (S3 Fig) and the *PFLU1130/1/2* operon but also include up-regulation of many other genes. In particular, we saw significant up-regulation for 54% of all RpoN-EBP controlled genes in the alternative rewiring pathway (Fig 3). For comparison, 70% of RpoN-EBP controlled genes were up-regulated in the primary rewiring pathway (via NtrC) [37]. Flagellar genes account for 12% of all RpoN-EBP controlled genes, and while the native regulatory targets of PFLU1132 are unknown, it is unlikely that it natively regulates 54% of these. Similarly, the first-step motile mutant (Δ*fleQ ntrB*-T97P) reported in Taylor and colleagues [37] leads to up-regulation of genes known to be involved in nitrogen assimilation (as expected as mutations are located in the Ntr pathway); however, it also results in up-regulation of many genes with no known regulatory link to NtrC. The large percentages of the RpoN regulon induced by these specific kinase mutations, in both the primary and alternative rewiring pathways, suggest first-step mutations confer a state of modified regulatory activity to PFLU1132 resulting in more frequent non-cognate interactions across many RpoN-EBP controlled genes including the flagellar genes.

## Evolutionary rescue of motility through alternative rewiring pathway is significantly constrained

If alternative rewiring routes are available to natural selection—why have we not seen them utilised in the presence of the primary rewiring route? In the alternative rewiring pathway, we measured the time taken for swimming mutants to evolve and the strength of the evolved phenotype in comparison to the primary rewiring route. Time to emergence (the length of time taken for motility to evolve on the soft agar plate) was recorded for all replicate experimental lines of the Δ*fleQ*Δ*ntrC* ancestor, as well as Δ*fleQ* as a comparison. Strikingly, only 9.4% of replicate experimental lines (*n* = 160) of Δ*fleQ*Δ*ntrC* evolved within the 6-week assay cutoff

compared to 100% for the Δ*fleQ* background (*n* = 22). Initial first-step motile Δ*fleQ*Δ*ntrC* strains also evolved within an average of 18.7 days from the assay start (Fig 4A), significantly longer than Δ*fleQ* took to rescue motility (4.2 days, *P* = 0.002, Dunn test). After first-step *PFLU1131* mutations, second-step mutants evolved rapidly in Δ*fleQ*Δ*ntrC* lines within an average of 2.2 days (Fig 4A). This was faster than the second-step mutants in the Δ*fleQ* background, which took 3.3 days to emerge (*P* = 0.0017, Dunn test).

The greatly increased time and low frequency for rescue motility in an Δ*fleQ*Δ*ntrC* background may be reflective of a small pool of accessible PFLU1131 mutations that can trigger rewiring of PFLU1132, perhaps due to a small mutational target size. One replicate line gained a de novo mutation in the DNA mismatch-repair gene *mutS* (S1 File), resulting in a frameshift and probable loss of function [46]. This strain, along with strains derived from it, possessed large numbers of additional SNPs (>80) including an SNP in *PFLU1131*. Loss of function to *mutS* is known to enhance mutability in *Pseudomonads* [47], and its occurrence may increase access to motility-rescuing mutations in PFLU1131.

From a phenotypic perspective, the first-step mutations in the *PFLU1131* pathway in a Δ*fleQ*Δ*ntrC* strain provide a far poorer motility phenotype than the analogous first-step mutations in a Δ*fleQ* background (Fig 4B). In a race assay, *PFLU1131*-del15 mutants swam 0.31 mm for every 1 mm swam by the most common first-step mutant in the primary rewiring route (*P* = 0.000356, Wilcox test). One possible explanation for this poor motility is a lack of significant up-regulation for the flagellar filament subunit FliC (S3 Fig) in the *PFLU1131*-del15 mutant.

The *PFLU1131*-del15 mutant additionally does not confer a significant defect to growth in shaking LB broth compared to its ancestral strain (Fig 4C; average relative area under the growth curve (rAUGC) of 1.55 and 1.48, respectively; *P* = 0.1132, Dunn test) and grew significantly better than the most common first-step mutant in the primary rewiring route (rAUGC of 1; *P* = 0.0081, Dunn test). The poor motility phenotype and the lack of significant fitness cost associated with *PFLU1131*-del15 likely indicates that the PFLU1131/2 system is a weaker activator of the flagellar motility that incurs less severe pleiotropy compared to NtrBC.

These findings indicate that the alternative pathway is the far poorer option for rescuing flagellar motility and flagellar gene expression compared to the primary NtrBC pathway—motility rescue in the Δ*fleQ*Δ*ntrC* background took significantly longer to occur, was significantly less frequent, and provides a significantly poorer flagellar motility phenotype. This solution also conferred a lower pleiotropic fitness cost, which in of itself may provide benefit but likely reflects to low flagellar expression and weaker rewiring. However, it is not clear why this should be the case—both NtrC and PFLU1132 are FleQ-homologs and are gaining mutations to their cognate kinases that facilitate rewiring. This suggests that there are other factors that constrain the evolutionary innovation of PFLU1132.

## Motility enhancing second-step mutations suggest that alterations to global gene regulatory network can facilitate rewiring in the alternative pathway

Following the first-step mutations in *PFLU1131* that unlock flagellar motility, our motile isolates develop second-step mutations that act to boost motility speed (S5A Fig), often accompanied by significant pleiotropic fitness costs for growth in LB broth (S5B Fig). These second-step mutations offer clues to the nature of constraining factors limiting innovation through rewiring of PFLU1132, as they represent evolutionary solutions to the poor motility provided to the first-step mutations.

Whole genome resequencing identified a diverse set of motility enhancing second-step mutations occurring at both a local (*PFLU1131/2* locus) and a global regulatory scale (Fig 5A;

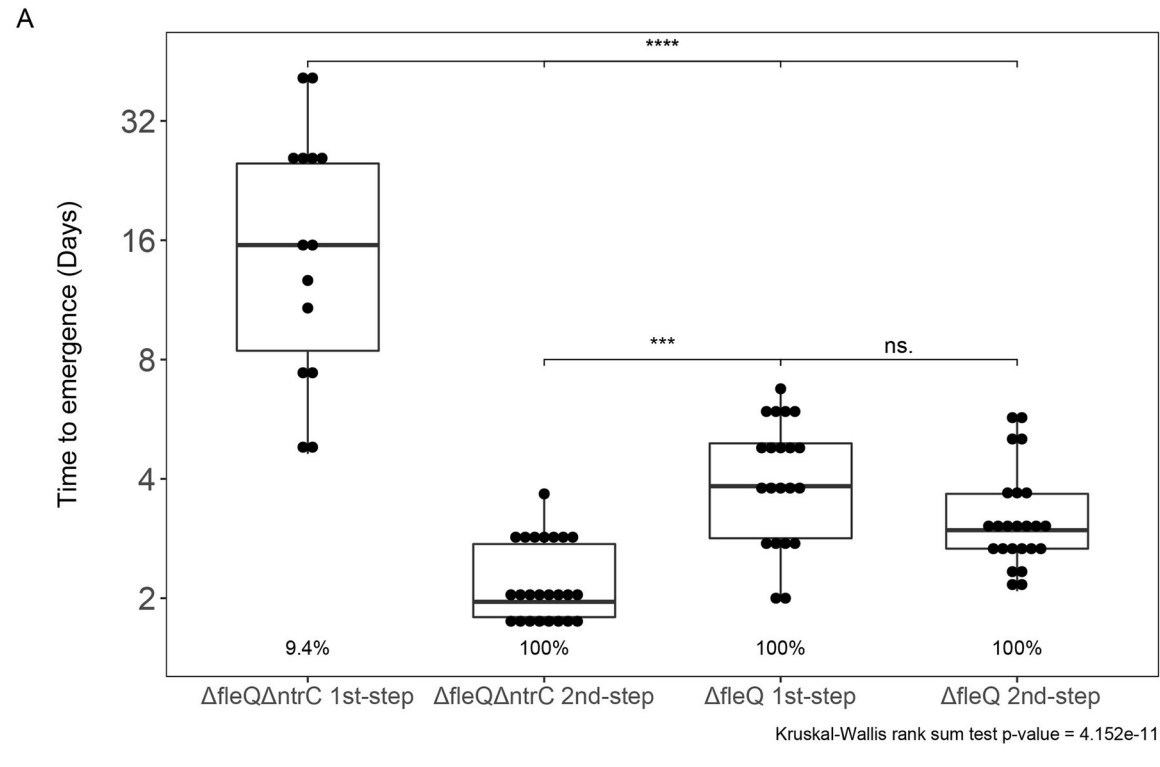

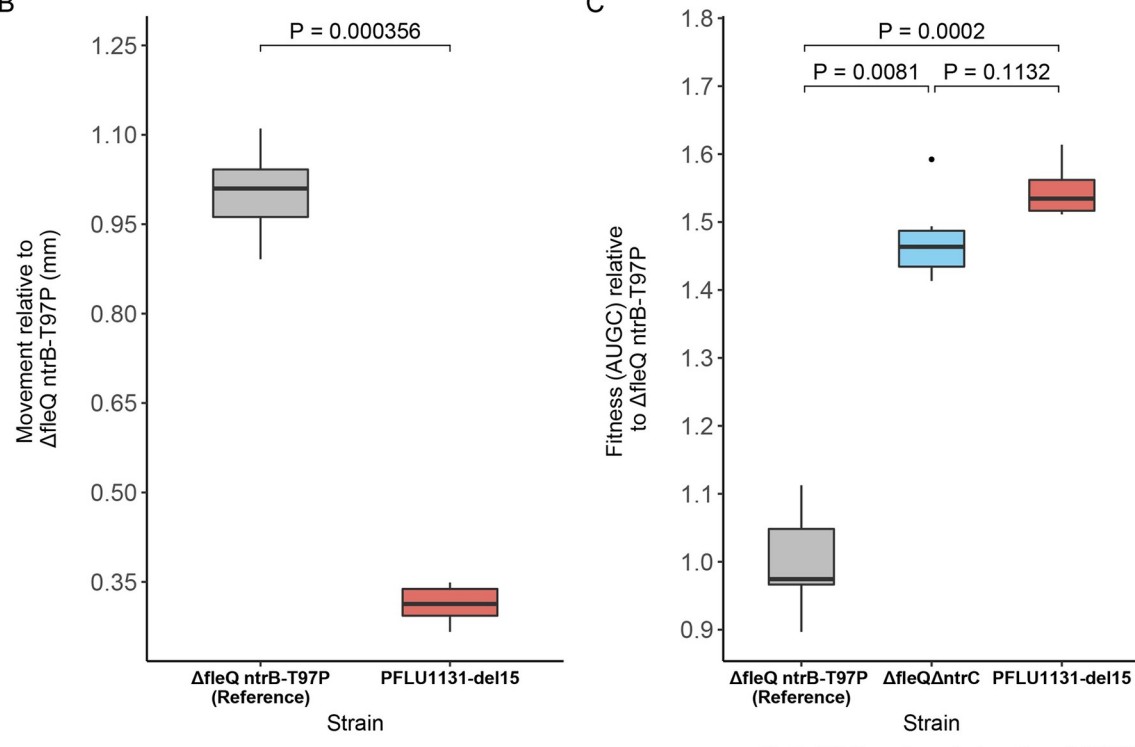

**Fig 4. Motility rescue through the alternative pathway is significantly slower and provides a poorer motility phenotype that displays lower pleiotropic costs. (A)** Time to emergence of motile zone (days) for first- and second-step mutants in the Δ*fleQ*Δ*ntrC* and Δ*fleQ* backgrounds. Percentages beneath each plot indicate proportion of replicate evolving that step within the 6-week assay cutoff. Significant differences in mean time to emergence are indicated as follows: $P < 0.0001 = ****$, $P < 0.001 = ***$, Ns. $= p > 0.05$ (Dunn test). **(B)** Race assay as measure of motility fitness. Distance moved over 24 hours in 0.25% agar LB plates measured relative to the Δ*fleQ*

*ntrB*-T97P mutant. *P* value indicated is generated from a two-sample Wilcox test. (**C**) Fitness in LB broth measured as area under the 24-hour growth curve (AUGC) relative to Δ*fleQ ntrB*-T97P. *P* values above plots generated from Dunn tests. AUGC values and growth curve plots are provided in S4 Fig. In parts B and C, *PFLU1131*-del15 is present in a Δ*fleQ*Δ*ntrC* genetic background. For all boxplots: box represents first to third quartile range; middle line represents median value; whiskers range from quartiles to maxima and minima. Data underlying parts A, B, and C can be found in S8, S9, and S10 Files, respectively.

full details of all mutations provided in S1 File, second step *n* = 18 (several first step lineages generates multiple second-step zones)). In contrast to the frequent second-step mutations observed in the HTH DNA-binding domain of NtrC [37], no analogous mutations were observed in the same domain of PFLU1132. Mutations that did occur in the *PFLU1132* gene were parallel and identified in only 2 lines, impacting the receiver domain of the transcription factor, which may further modify activity of the PFLU1132 regulator by altering the interactions with its kinase. One line gained a secondary *PFLU1131* mutation, an SNP in the HAT-Pase domain along with the first-step SNP A547D already present (Fig 2B), which may further boost kinase interactions. A mutation to the promoter region of the *PFLU1131/2* operon was also observed.

Other second-step mutations can be grouped into 3 other broad categories. (i) The first are mutations in the operon *PFLU1583*/4, accounting for 24% of second-step mutations (Fig 5A, purple). This gene pair encodes a putative anti-sigma factor PP2C-like phosphatase and a putative STAS family anti-sigma factor antagonist (*PFLU1583* and *PFLU1584*, respectively). Mutations in *PFLU1583* were generally loss of function suggesting loss of repression on an unknown sigma factor. This may act on RpoN, the partner sigma factor of PFLU1132, in regulating gene expression; however, this cannot be tested easily through *rpoN* knockout, and *P. fluorescens* SBW25 encodes approximately 31 other putative sigma factors [45], which may instead be the mechanistic targets of this mutation. Transcriptomic analysis of a *PFLU1583* mutant (*PFLU1583*Δ48–74) identified sigma factor RpoE as up-regulated; however, *rpoE* knockout did not negate the motility enhancing effect of the mutation (S6A Fig). To test that PFLU1583/4 are not acting to enhance motility through another RpoN-EBP, we engineered *PFLU1583*Δ48–74 in a Δ*fleQ*Δ*ntrC* background in the absence of a *PFLU1131* mutation. This strain was immotile, indicating that these anti-sigma factor mutations depend on the function of the PFLU1131/2 system to confer a swimming phenotype (S6B Fig). (ii) The second category of mutations impact core gene expression components that will affect global GRN function (Fig 5A, green). These included mutations to *rho*, *rpsK*, and *rpoC* that encode core gene expression machinery and will impact expression of most genes in the genome. (iii) The final category of motility enhancing mutations fit a metabolic theme (Fig 5A, blue). These mutations occur in genes involved in central carbon metabolism, which will likely have a significant impact on global gene expression through alterations to central carbon flux, likely incurring pleiotropic fitness trade-offs [48], and may act on RpoN indirectly—for example, via core carbon metabolic regulator CbrB (an RpoN-EBP)—or signal to the PFLU1131 sensor kinase via internal metabolic flux.

The diversity and nature of these second-step mutations suggest global regulatory strategies to facilitate flagellar gene expression through the alternative rewiring pathway. To understand their regulatory impact, transcriptome analysis was performed on a pair of representative mutants: the most common second-step mutant (anti-sigma factor mutant, *PFLU1583* Δ48–74), which represented a mutation with global regulatory effects, and the *PFLU1131/2* promoter mutation as a representative of mutation with predicted "local" regulatory effects. Together, these mutations represent the 2 loci that constitute 56% of identified second-step mutations.

Principal component analysis indicates that the transcriptomic profile of the anti-sigma factor mutant differs significantly from the profile of the *PFLU1131/2* promoter mutation. Both

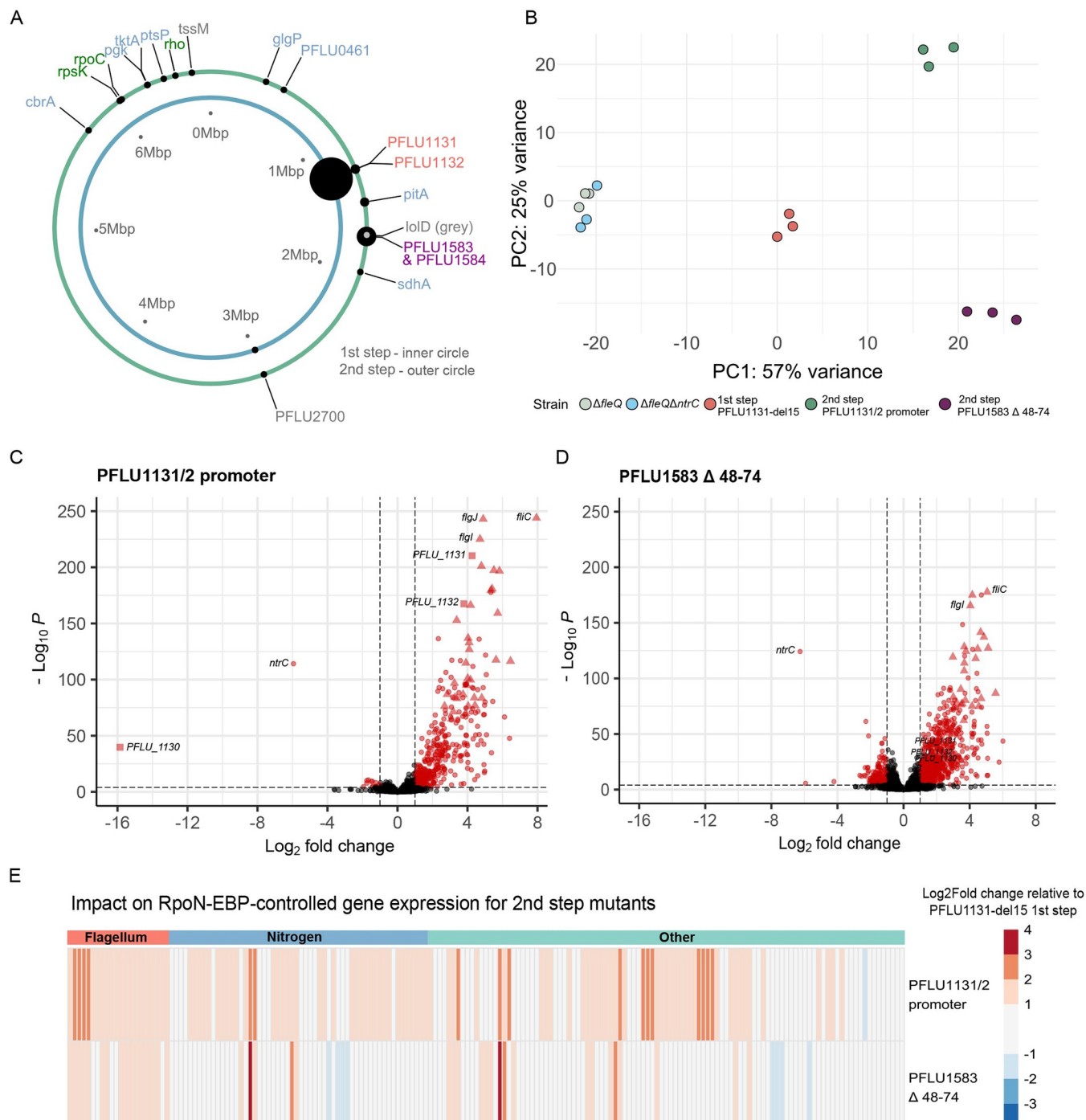

**Fig 5. Diverse second-step mutations indicate multiple global regulatory strategies for enhancing rewiring through PFLU1132. (A)** Diagram of mutational distribution across the SBW25 chromosome for first- and second-step motility mutants. Each ring represents the chromosome, with black dots indicating non-synonymous mutations. The size of each dot is proportional to the number of mutations occurring in that locus in independent replicates. Mutational target names are coloured by functional category: red: *PFLU1131/2*; purple: *PFLU1583/4*; blue: metabolic; green: global regulatory; grey: other. **(B)** Principal component analysis of transcriptomic data for ancestral, first-step, and second-step motility mutants. **(C and D)** Volcano plots of differentially expressed genes for second-step mutations relative to Δ*fleQ* ancestor (mutants are themselves Δ*fleQ*Δ*ntrC* backgrounds). Red points are significantly differentially expressed. Triangles indicate flagellar genes; squares indicate *PFLU1131/2* and adjacent genes; and circles indicate all other genes. Vertical and horizontal dashed lines indicate significance cutoff values on both the *x* and *y* axes. **(E)** Log2Fold change in gene expression for RpoN-EBP controlled genes after *PFLU1583* Δ48–74 mutation—relative to the first-step *PFLU1131*-del15 mutant (mutants are themselves Δ*fleQ*Δ*ntrC* backgrounds). RpoN-EBP controlled genes grouped by function are indicated by the coloured bars above the plot: red: flagellum; blue: nitrogen; teal: other/unknown. Data underlying parts B, C, and D can be found in S6 File, and part E in S11 File.

mutations resulted in similar variation across PC1 relative to the first-step *PFLU1131*-del15 mutant, but opposite variation in PC2 (Fig 5B). Both the *PFLU1131/2* promoter and anti-sigma factor mutations result in net up-regulatory effects on the transcriptome reflected by positive skewed volcano plots (Fig 5C and 5D) albeit with differing expression patterns. Both mutations have the effect of further up-regulating RpoN-EBP controlled genes with 22% and 60% being up-regulated for the anti-sigma factor mutant and the *PFLU1131/2* promoter mutant, respectively (Fig 5E).

In general, for the alternative rewiring pathway, genomic and transcriptomic analysis of second-step mutations indicate enhancement of rewiring through the PFLU1132 transcription factor, albeit utilising differing (local and global) mechanisms. These results highlight that diverse mutations with both targeted and global regulatory effects can help facilitate transcription factor rewiring.

## Second-step promoter capture mutation suggests importance of transcription factor expression for rewiring

Although second-step mutations were diverse in the alternative rewiring pathway, one promoter capture event in particular provides evidence for the role of increased expression of the rewired transcription factor in strengthening the motility phenotype. To gain capacity to drive expression of non-cognate genes, the PFLU1132 transcription factor will need to first saturate its native regulatory interactions before it can engage in low-affinity interactions with FleQ-controlled genes. High expression of active transcription factor can provide these conditions, elevating the concentration of transcription factor in the cell to permit greater expression of flagellar genes [49,50].

While many of the second-step mutations detailed above may impact *PFLU1131/2* expression indirectly (RpoC, Rho), the *PFLU1131/2* promoter mutation directly effects expression of the two-component system. This mutation is a 1.59-kbp deletion resulting in total loss of the *PFLU1130* gene, and in *PFLU1131/2* becoming part of the *PFLU1127/8/9* operon (Fig 6A). Upstream of this new combined operon sits a predicted RpoN binding site [44], so this deletion positions *PFLU1132* in a new operon under the control of RpoN, which may create a positive feedback loop where this RpoN-dependent regulator drives its own expression—either through rewired regulation or native control of this promoter. This genetic rearrangement significantly up-regulates both *PFLU1131* and *PFLU1132* with 4.26 and 3.79 Log2Fold increases in expression compared to the non-motile Δ*fleQ* strain, respectively (Fig 6B, raw values for this and other tested strains provided, S5 File). Up-regulation of this two-component system corresponds with the 60% increase in expression of RpoN-EBP controlled genes discussed above (Fig 5E), indicating an increase in the strength of PFLU1132 rewiring—inducing greater flagellar gene expression (S3 Fig). The 1.59-kbp deletion results in loss of an intergenic region that typically separates the 2 combined operons. This region is predicted to contain a rho-dependent terminator (S3 File), which may typically prevent transcriptional readthrough into *PFLU1130/1/2*. Such a terminator will restrict *PFLU1131/2* expression via readthrough and constrain its ability to achieve high concentrations that may facilitate rewiring. The *rho* mutation observed in another second-step motile strain may also act to increase readthrough at this site. In comparison to this promoter mutation, the other second-step mutation tested (anti-sigma factor *PFLU1583* Δ48–74) does not significantly impact *PFLU1131/2* expression (Fig 6B), so this mutation likely influences rewiring of the *PFLU1132* transcription factor through a separate mechanism.

In summary, expression of *PFLU1131/2* appears to be a major factor constraining rewiring of this regulatory system and rescue of flagellar motility, and overcoming this constraint facilitates evolutionary innovation through this alternative rewiring route.

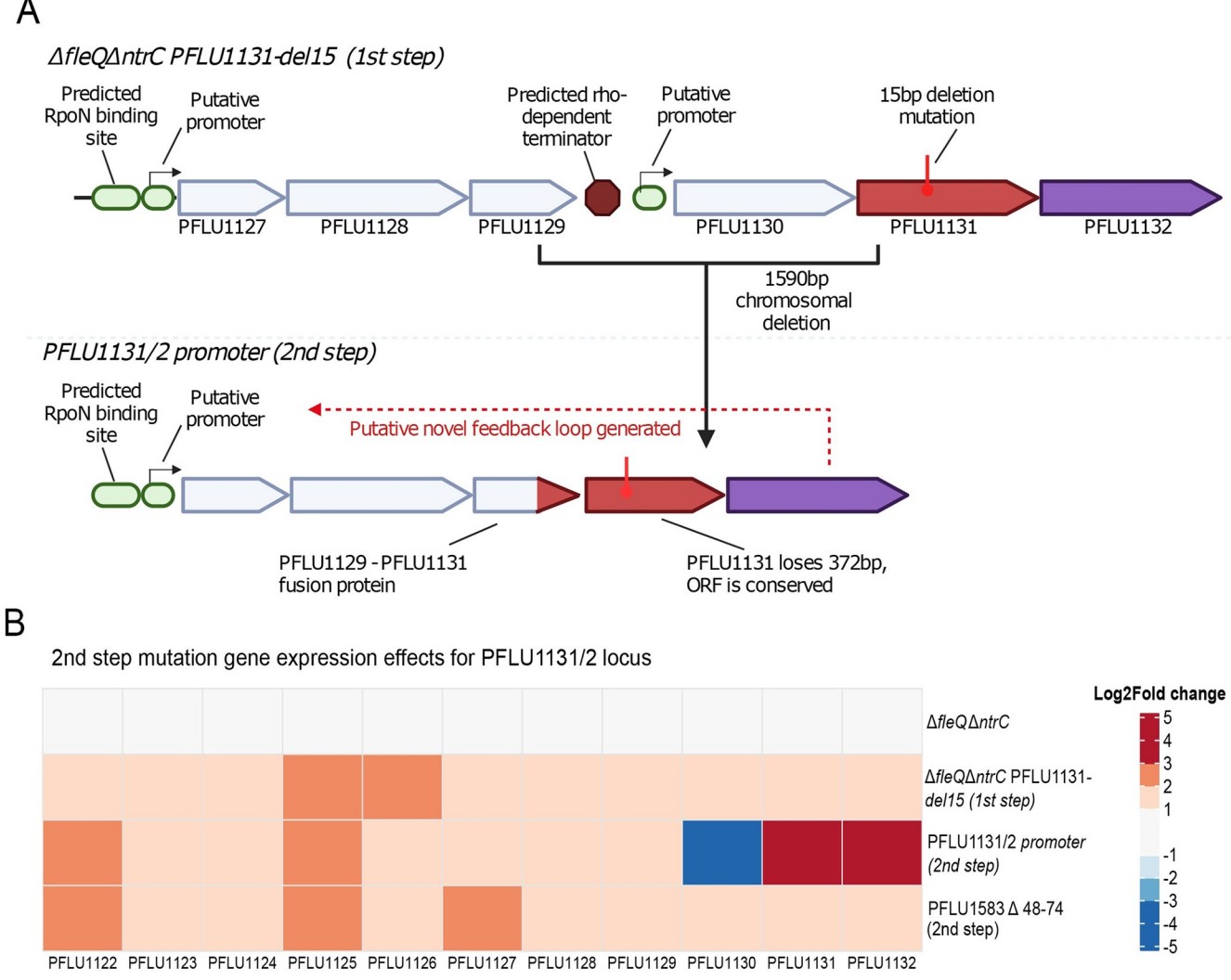

**Fig 6. Second-step *PFLU1131/2* promoter mutation increases *PFLU1131/2* expression. (A)** *PFLU1130/1/2* genetic locus in the SBW25 chromosome, and impact of the 1.59-kbp deletion identified in 1 second-step motile mutant. Location of the RpoN binding site upstream of PFLU1127/8/9 was predicted by Jones and colleagues [44]. The presence of the rho-dependent terminator was determined by use of RhoTermPredict [80]. **(B)** Log2Fold change in gene expression (relative to Δ*fleQ* ancestor) heatmap of *PFLU1130/1/2* and nearby RpoN-EBP controlled genes, including *PFLU1127/8/9* in the ancestral Δ*fleQ*Δ*ntrC* strain, first- and second-step motility mutants. Data underlying part B of this figure can be found in S12 File.

### Increasing transcription factor expression in motile mutants results in faster motility phenotypes

To test the importance of transcription factor gene expression for rewiring, we made use of the complementation strains presented in Figs 2C and S1. These experiments produced strains where the primary (*ntrC*) and alternative (*PFLU1132*) rewired transcription factor genes are deleted from their native loci and reintroduced on an L-rhamnose titratable promoter system. This was done in the presence of their respective kinase mutations that conferred first-step slow spreading motility in each rewiring pathway. Concentration of L-rhamnose added to the media positively correlated with distance moved (Fig 7A) for both the primary rewiring pathway (i.e., NtrC expression system; ρ = 0.984, Spearman test) and the alternative rewiring pathway (i.e., PFLU1132 expression system; ρ = 0.865, Spearman test). We confirmed increasing L-

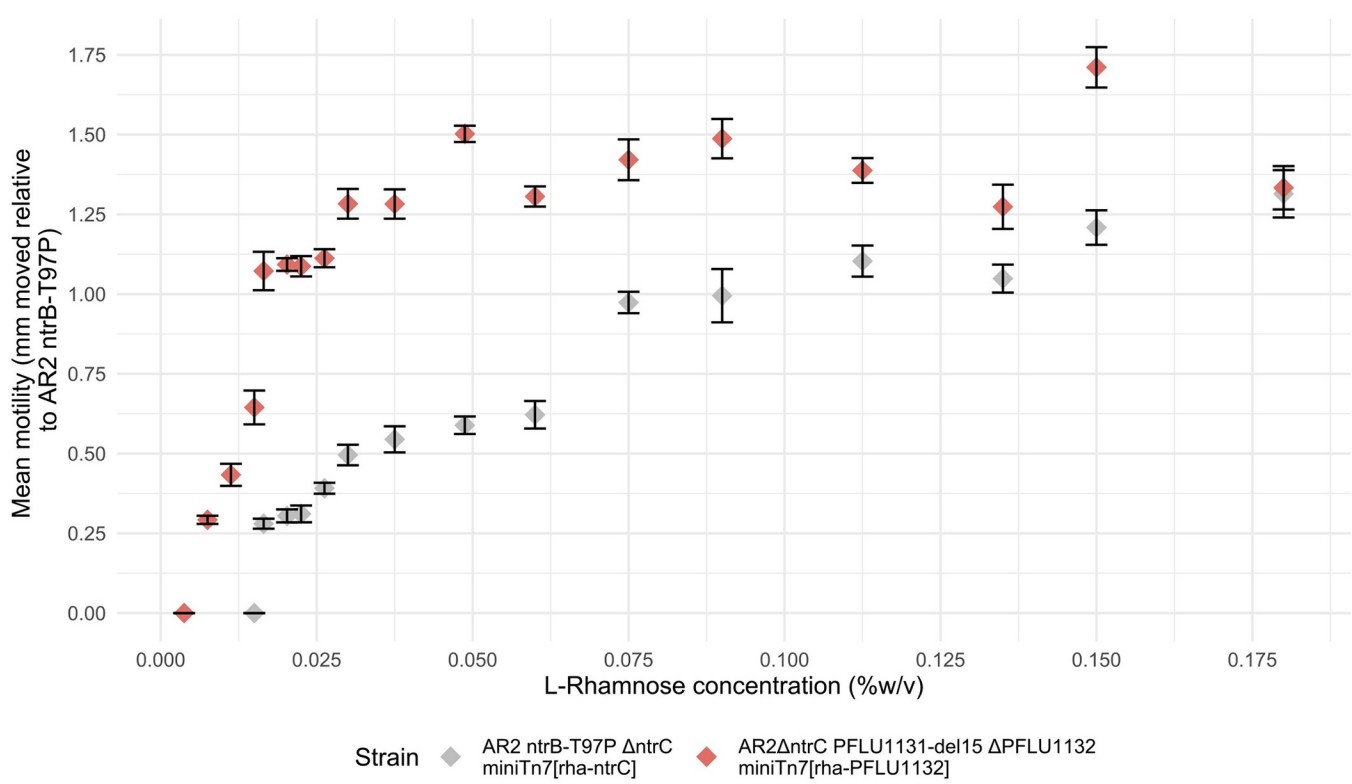

Strain: AR2 ntrB-T97P ΔntrC miniTn7[rha-ntrC] | AR2ΔntrC PFLU1131-del15 ΔPFLU1132 miniTn7[rha-PFLU1132]

Spearman's rho: rha-ntrC = 0.984, rha-PFLU1132 = 0.865

**Fig 7. Impact of RpoN-EBP gene expression on motility speed. (A)** Mean motility speed (mm relative to Δ*fleQ ntrB*-T97P) plotted against increasing L-Rhamnose concentration (%w/v). Mean of 6 biological replicates for each point. First-step motility mutants Δ*fleQ ntrB*-T97P and Δ*fleQΔntrC PFLU1131*-del15 had their respective RpoN-EBPs knockout out and reintroduced on a miniTn7 transposon with the RpoN-EBP under control of the PrhaBAD and *rhaSR* rhamnose-inducible expression system. In full, these were Δ*fleQ* ntrB-T97P Δ*ntrC* miniTn7[*rhaSR*-PrhaBAD-stRBS-ntrC] (rha-*ntrC*) and Δ*fleQΔntrC PFLU1131*-del15 ΔPFLU1132 miniTn7[*rhaSR*-PrhaBAD-stRBS -PFLU1132] (rha-1132). **(B)** Motility speed (mm relative to Δ*fleQ ntrB*-T97P) of common pathway ntrB-T97P first-step mutant, *PFLU1131*-del15 first-step and PFLU1131/2 promoter second-step mutants. Whiskers represent standard deviation above and below the mean value. Data underlying parts A and B of this figure can be found in S13 and S14 Files, respectively.

rhamnose concentration increased expression from the L-rhamnose titratable system (RhaSR-PrhaBAD) in our bacterial strain backgrounds by testing LacZ activity with increasing L-rhamnose concentration for *lacZ* under control of the L-rhamnose titratable promoter construct in the Δ*fleQ* and Δ*fleQΔntrC* backgrounds. In both cases, LacZ activity positively correlated with L-rhamnose concentration (ρ = 0.981 and ρ = 0.955, respectively, Spearman tests), indicating that increasing L-rhamnose concentration results in increasing expression of the gene introduced in the *rhaSR-PrhaBAD* expression system (S7 Fig).

This allows us to conclude that increasing expression of *ntrC* and *PFLU1132* in the presence of their first-step kinase mutations granted stronger motility phenotypes, facilitating their rewiring. The expression level of a transcription factor can therefore significantly impact its propensity for rewiring, and more highly expressed transcription factor may be expected to engage in non-canonical regulatory interactions more readily [51].

## Discussion

Essential to understanding why some organisms adapt faster than others in the face of environmental challenges is knowledge of how GRNs and regulatory systems evolve. Evolutionary innovation of transcription factors within these networks depends on gain-rewired

connections; however, the rules of when and where these novel connections develop within the context of the GRN are currently unknown. Using our microbial model system, we have identified 2 available pathways for rescuing flagellar motility in a maladapted Δ*fleQ* network through mutations to the NtrBC and PFLU1131/2 two-component systems. However, one pathway—through NtrBC—is always used to the exclusion of the other when both are present. By comparing these available evolutionary pathways, we can resolve why NtrC is more accessible for evolutionary rewiring due to its position within its GRN. In addition, we can identify 3 key properties that facilitate transcription factor rewiring and therefore opportunities for evolutionary innovation.

Firstly, our results confirm the importance of shared structural homology between the absent and recruited transcription factor. In our system, the shared structural homology of NtrC and PFLU1132 with FleQ grants a basal affinity for regulating flagellar genes that evolution can act upon. This likely explains why structurally unrelated transcription factors do not rewire in our model and indicates that similarity to FleQ facilitates the rewiring observed in NtrC and PFLU1132. The RpoN-EBP family was likely generated through rounds of ancestral gene duplications, indicating that rewiring within a family of regulators with a shared ancestry may be more likely to occur. However, it is not the case that the regulators that are most structurally homologous to FleQ are the first to be repurposed [39], so homology is not the deciding factor in which RpoN-EBPs rewire to rescue flagellar motility. It is possible that the RpoN-EBPs represent a hyper-evolvable family of transcription factors through the nature of their connectivity and mechanism of transcriptional regulation via RpoN. To investigate this evolutionary rescue of a non-RpoN-dependent trait would need to be tested using a different maladapted GRN rescue model—as such, a limitation of our findings is that they principally apply to situations where a rewired transcription factor is homologous to that which it replaced. Secondly, hyperactivation of the co-opted transcription factor benefits gain of rewired regulatory activity. In our model system, first-step mutations that granted a slow motile phenotype were always achieved through mutation of a kinase, resulting in hyperactivation of their cognate transcription factors. Most transcription factors are subject to some form of posttranslational control, most commonly through phosphorylation of a REC receiver domain in bacteria [52]. Novel regulatory action may occur transiently or only briefly, so ensuring that the transcription factor is constantly activated increases the likelihood of these non-cognate regulatory events leading to altered gene expression. Finally—and perhaps most importantly—high gene expression of the transcription factor boosts rewired activity. High expression of a transcription factor can lead to saturation of native binding sites. Any unbound, active transcription factor is then free to engage in rewired activity. These low-affinity interactions will occur more readily if there is a higher concentration of transcription factor present, following laws of mass action kinetics [53]. Typically, transcription factors are expressed at low levels in bacteria [54], due to only a small protein copy number being needed for a small number of binding sites in most cases [55]. Transcription factor expression level may well be optimised at a low level to prevent misregulation. Indeed, highly expressed and connected transcription factors were found to result in greater network perturbations when synthetically rewired [19], which may reflect a greater ability to establish novel regulatory connections. Many molecular biology studies use overexpression/overactivation of proteins of interest as a means to determining their biological functions [56]. Our findings also suggest that this approach should perhaps be used more cautiously, certainly when studying transcription factor function.

Aside from mutations directly targeting activation and expression of *PFLU1131/2*, we observed several distinct strategies that enhance PFLU1132 rewiring through global regulator changes. These strategies likely impact the wider GRN and transcriptome significantly, with mutations to RpoC, RpsK, and Rho being particularly striking. RpoC and RpsK mutations

may act to increase expression of most genes, including *PFLU1131/2* and the flagellar genes. Mutations to the *PFLU1583/4* anti-sigma factor system also indicated a strategy to boost rewiring that would have a significant impact of the wider GRN. Global regulators are often sites of mutation and innovation [57,58], with mutations affecting sigma factor function in particular having large impacts on global gene expression, which can drive phenotypic innovation [59]. Changes to sigma factors have been shown to facilitate GRN remodelling, as was the case for a study of *Pseudomonas aeruginosa* during adaptation to the cystic fibrosis lung [13]. Our study highlights that such mutational strategies can also help facilitate rewiring and innovation through specific transcription factors. Each of these mutational strategies that enhanced rewiring in PFLU1132 acted to alter the preexisting GRN architecture around the transcription factor, whether at a local level by altering the *PFLU1131/2* promoter or at a global regulatory level.

From the evidence outlined, preexisting GRN architecture and local sequence context can be a key constraining factor for the rewiring of a transcription factor. For example, the ease by which a transcription factor can gain high expression depends on the GRN structure it sits within. Whether a transcription factor exists within a two-component system or acts without a cognate kinase as a one-component system could influence the ease by which it is activated and rewired. Additionally, many transcription factors promote their own expression [60] (positive autoregulation) and can generate high self-expression through runaway positive feedback. Such positive feedback loops are typically contained by a negative repressor [61–63], including in the case of NtrBC (Fig 1B). Mutational loss of repression can therefore be an evolutionarily accessible mechanism to achieve high expression levels of a transcription factor. In our model system, GlnK prevents runaway *ntrBC* autoregulation by inhibiting NtrB [64] (Fig 1B). All previously observed NtrBC pathway mutations were single de novo mutations that act to remove repression through GlnK, both directly or indirectly [37,38,41]. These have a dual effect, both hyperactivating the NtrBC system and strongly up-regulating *ntrBC* expression through their positive autoregulatory loop. In contrast, we have identified in this study that to achieve the same effect of high activity and high expression in PFLU1131/2, multiple mutations were required (Fig 1C). This highlights the clear evolutionary advantage NtrBC has for being rewired compared to PFLU1131/2.

From the evidence outlined, preexisting GRN motifs can be predicted to significantly bias the evolvability of the transcription factors under their control. This is strongly supported by the literature: Mutational loss of a negative repressor leading to overactivity and overexpression of a positively autoregulated transcription factor has been observed to drive adaptation in several experimental and clinical settings [65–67].

Considering the alternative scenario, a transcription factor that represses its own expression (negative autoregulation) will hinder the gain of a high expression level, due to the nature of self-repression. Interestingly, the hierarchy favouring NtrBC over PFLU1132 matches previous studies of evolutionary pathway hierarchies [34], where mutational target size predicts that loss of repressors will be favoured over other precise mutations. Additionally, studies of compensation after loss of GRN "hub" genes have also found such compensation to be highly repeatable, and to offer a diverse array of phenotypes after compensation, potentially opening additional evolutionary paths [68]. Our study highlights that in the context of GRN structures, connectivity to other network components can constrain or enable innovation through rewiring, alongside other effects including mutational target size. Rewiring can also occur through mutation to *cis*-regulatory elements (promoters, terminators, binding sites) [69]; however, we observed a limited role for *cis*-regulatory mutations in our model, with the only promoter mutations targeting the rewired transcription factor itself. This may be due to the large number of flagellar gene promoters that would need to mutate to gain a regulatory connection to PFLU1132, constituting a significant mutational target size [40].

Our study provides empirical evidence for mechanisms and evolutionary drivers of regulatory evolution. GRNs are key to the adaptability of organisms, and our results reveal the genetic regulatory factors that generate evolvable control systems. We find that feedback loops that can generate high expression of regulators after a small number of mutations can facilitate rapid innovation and adaptation. Previously, we have hypothesised that some transcription factors may be 'primed" for evolutionary innovation by virtue of the GRN architecture they exist within [31]. Our data support this by suggesting that transcription factors under positive feedback regulatory structures may more readily rewire. These findings can also be utilised in the design of genetic regulatory systems to have reduced evolvability, a key future goal for synthetic biology projects where the evolutionary change of a constructed system is detrimental [70,71]. Designing systems that avoid easily accessible routes to hyperactivation and hyperexpression of their components could help prevent evolutionary rewiring of engineered GRNs. Preexisting GRN structure can be expected to significantly bias patterns of diversification in regulatory systems and subsequently shape genomic evolution by influencing the speed in which genetic circuitry can adapt to changing conditions. This is highly important for populations adapting during sudden changes such as niche transitions including the emergence of new pathogens. Our data move us closer to the eventual goal of empirically building a set of principles by which GRNs adapt and offers unique insights into how these systems can function and influence the evolution of their components.

## Materials and methods

### Strains and culture conditions

All ancestral lines in this study are derived from *P. fluorescens* SBW25Δ*fleQ* IS-ΩKm-hah: PFLU2552 (referred to throughout as Δ*fleQ*). Removal of the flagellum master regulator FleQ and transposon-insertional disruption of the gene *viscB* (PFLU2552) renders this strain immotile as detailed previously [37,72]. All strains were cultured on LB (Miller) media at 27°C. *Escherichia coli* strains for cloning were cultured on LB media at 37°C. Strains, plasmids, media supplements, and primers are detailed in S4 File.

### Knockout mutant construction by two-step allelic exchange

Knockouts of *ntrC* and *PFLU1132* were achieved using two-step allelic exchange following the protocol of [73] with some alterations. In brief, 400-bp flanking regions up- and downstream of the target gene were amplified and joined by strand overlap extension (SOE) PCR to create a knockout allele. This was inserted into the allelic exchange suicide vector pTS1 (containing *sacB* and *tetR*) by SOE cloning [74] to create the knockout plasmid and transformed into the conjugal *E. coli* strain ST18 by chemical competence heat shock. Knockout plasmids were transferred from *E. coli* ST18 to Δ*fleQ* by two-parent puddle mating and merodiploids selected for on LB supplemented with kanamycin sulphate and tetracycline hydrochloride but lacking 5-ALA supplement required for growth of the *E. coli* ST18 auxotrophic Δ*hemA* mutant. Merodiploids were cultured overnight in LB lacking tetracycline selection and diluted before spread plating onto NSLB media supplemented with 15% w/v sucrose for *sacB* levansucrase-mediated counterselection of the pTS1 plasmid backbone. Sucrose-resistant colonies were isolated and screened for tetracycline sensitivity. Chromosomal presence of the knockout allele and absence of target gene coding sequences was confirmed by colony PCR. Knockout of *ntrC* left an ORF encoding an 11 amino acid truncated protein, and knockout of *PFLU1132* left an ORF encoding an 8 amino acid truncated protein. All other genomic features including operon structure and terminator regions were left intact.

## Motility evolution experiments

$\Delta fleQ\Delta ntrC$ and $\Delta fleQ\Delta ntrC\Delta PFLU1132$ were challenged to rescue motility in the absence of the FleQ master flagellar regulator on soft agar, as described previously [37,38]. Pure colonies were picked and inoculated into 0.25% agar LB plates made as described in Alsohim and colleagues and incubated at 27°C. Plates were checked a minimum of twice daily for motility, recording time to emergence. Motility was identified through visual inspection of plates—flagellar motility is visually distinct from other motility modes in *P. fluorescens*, characterised by semicircular zone of growth expanding from the central growth mass and moving through the entire depth of the agar layer [72]. Motile zones were sampled immediately and always from the leading edge. Motile isolates were streaked on LB agar, and a pure colony picked and stored at −80°C as glycerol stocks of LB overnight cultures. All subsequent analysis was conducted on these pure motile isolates. Experiment was run for 6 weeks and any replicates without motility after this cutoff recorded as having not evolved.

## Bacterial growth and motility fitness assays

The fitness of the motility phenotype of evolved $\Delta fleQ\Delta ntrC$ strains was tested by measuring distance moved over 24 hours of incubation in 0.25% agar LB plates. Six biological replicates of each strain were grown as separate overnight cultures. Cultures were adjusted to an $OD_{595} = 1$ and resuspended in PBS. Soft agar plates were inoculated with 1 μL of these suspensions by piercing the surface of the plate with the pipette tip and then effusing the sample into the gap left by the tip. Plates were incubated for 24 hours at 27°C, and photographs taken of motile zones. Surface area moved was then calculated from the radius of the concentric motile zone measured from these images ($A = \pi r^2$). Values were square root transformed before plotting.

Growth in shaking LB broth was measured by inoculating 99 μL of sterile LB broth with 1 μL of the $OD_{595} = 1$ PBS cell suspensions for each replicate in a 96-well plate. Plates were incubated at 27°C with 180 rpm shaking in a plate reader, recording $OD_{595}$ every hour for 24 hours. Area under the bacterial growth curve was calculated using the growthcurver package in R and plotted [75]. Area under the bacterial growth curve provides a metric for fitness, as it accounts for the characteristics of lag- and log-phase growth, as well as the final carrying capacity of the population. All growth curves were plotted and inspected prior to calculation of AUGC to ensure similar AUGC values corresponded to similar curve shapes (S3 Fig).

## Mutation identification by whole genome resequencing and PCR amplicon Sanger sequencing

To identify motility rescuing mutations, genomic DNA was extracted from motile strains and their ancestral strain using the Thermo Scientific GeneJET Genomic DNA Purification Kit. Genomic DNA was quality checked using BR dsDNA Qubit spectrophotometry to determine concentration and nanodrop spectrophotometry to determine purity. Illumina HiSeq sequencing was provided by MicrobesNG (Birmingham, UK), with a minimum 30× coverage and a sample average of 114× coverage. Returned paired-end reads were aligned to the *P. fluorescens* SBW25 reference genome [45] using the Galaxy platform [76]. INDELs were identified using the integrated genomics viewer [77]. For SNP identification, the variant calling software SNIPPY was used with default parameters [78]. Protein domains affected by mutations detailed in this article were predicted from amino acid sequences using pfam, SMART, and BLASTp. For a subset of additional motile mutants, PCR amplification and subsequent Sanger sequencing of *PFLU1131* were performed as this gene was mutated in all previously sequenced strains. These were done using the service provided by Eurofins Genomics. PCR amplicons

were purified using the Monarch PCR & DNA Cleanup Kit (New England Biolabs). Mutations were identified by alignment of the returned PFLU1131 sequence against the *P. fluorescens* SBW25 reference genome using NCBI BLAST.

Three second-step isolates derived from line 24 were sampled due to observed colony morphology variation. After whole genome resequencing, it became apparent that this variation was due to the presence of a *mutS* mutation and not due to diverse motility strategies. These isolates all have the mutation *PFLU1132*-D111G (S1 File, isolate IDs 24-S1, 24-LCV, and 24-SCV); however, this is likely not due to parallel evolution, but instead a single instance of a *PFLU1132*-D111G mutant that then diversified due to *mutS* mutation. For all subsequent analysis, these 3 isolates were treated as a single instance of *PFLU1132*-D111G.

### RNA sequencing

Whole-cell RNA was extracted from 20 OD units of *P. fluorescens* cultures in mid-log phase growth ($OD_{595}$ approximately 1.5). Cultures were incubated in LB at 27˚C and 180 rpm shaking. Extractions were performed for biological triplicates of each strain of interest. Upon reaching the desired OD, growth and RNA expression were halted by addition of a ½ culture volume of ice-cold killing buffer (20 mM $NaN_3$, 20 mM Tris–HCl, 5 mM $MgCl_2$). Cells were pelleted, and the killing buffer removed. A lysis buffer of β-mercaptoethanol in buffer RLT from the Qiagen RNeasy extraction kit was used to resuspend pellets, which were then lysed by bead milling at 4,500 rpm for 45 seconds with lysing matrix B. Lysates were spun through columns from the Qiagen RNeasy extraction kit, and the extraction completed following the RNeasy kit protocol. A DNase I treatment step was included between washes with buffer RW1 by adding DNase I directly to the column from the RNase-free DNase kit (Qiagen) following kit protocol. Samples were eluted in nuclease-free water and subsequently treated with TURBO DNase from the Turbo DNA-free kit (Invitrogen) following kit protocols.

Purified RNA concentration was measured by Qubit RNA BR assay (Thermo Scientific), RNA quality by nanodrop spectrophotometry, and RNA integrity by agarose gel electrophoresis. RNA sequencing was provided by Oxford Genomics (Oxford, UK). Samples were ribodepleted, and the mRNA fraction converted to cDNA with dUTP incorporated during second strand synthesis. The cDNA was end-repaired, A-tailed, and adapter-ligated. Prior to amplification, samples underwent uridine digestion. The prepared libraries were size selected, multiplexed, and QC'ed before paired end sequencing over one unit of a flow cell in a NovaSeq 6000. Returned data were quality checked before these were returned in fastq format. Reference-based transcript assembly was performed using the Rockhopper RNA sequencing analysis software [79] to generate transcript read counts. Differential gene expression analysis was performed using the DESeq2 package of the R statistical coding language to create lists of differentially expressed genes and corresponding Log2Fold changes in gene expression.

### Rhamnose-inducible expression system constructs

Complementation of *ntrC* and *PFLU1132* knockouts as well as inducible expression experiments were performed by single-copy chromosomal reintroduction of the RpoN-EBP genes under control of a *rhaSR-PrhaBAD* L-rhamnose–inducible expression system [80]. The RpoN-EBP ORF was amplified and a strong ribosome binding site (stRBS) introduced upstream after a 7-bp short spacer between the site and the start codon. This construct was placed downstream of the *PrhaBAD* promoter on the miniTn7 suicide-vector pJM220 (obtained from the Addgene plasmid repository, plasmid #110559) by restriction-ligation and transferred to *E. coli* DH5α by chemical competence heat shock. The miniTn7

transposon containing the *rhaSR* genes and the PrhaBAD-stRBS-RpoN-EBP construct were transferred to the *P. fluorescens* chromosome by transposonal insertion downstream of the *glmS* gene via four-parent puddle mating conjugation [81]. The relevant *E. coli* DH5α pJM220-derived plasmid donor was combined with recipient *P. fluorescens* Δ*fleQ* strains, transposition helper *E. coli* SM10 λpir pTNS2 and conjugation helper *E. coli* SP50 pRK2073, and gentamicin-resistant *Pseudomonas* selected for on LB supplemented with gentamicin sulphate and kanamycin sulphate. Chromosomal insertion of the correct miniTn7 transposon was confirmed by colony PCR.

### β-galactosidase activity assay

To validate the expression dynamics of the rhaSR-PrhaBAD system in the genetic backgrounds used in this work, the same system described above was introduced into *P. fluorescens* Δ*fleQ* and Δ*fleQ*Δ*ntrC* with *lacZ* under the control of PrhaBAD. This was obtained from the pJM230 plasmid [80] (obtained from the Addgene plasmid repository, plasmid #110560). Overnight cultures of each strain were set up in biological triplicate and used to inoculate 9 mL cultures of LB supplemented with a range of L-rhamnose concentrations at a bacterial $OD_{600} = 0.05$. Cultures were incubated shaking 180 rpm at 27˚C until they reached an $OD_{600} = 0.5$; 2 mL was spun down and frozen at −20˚C until needed. Cell pellets were resuspended in Z-buffer (60 mM $Na_2HPO_4$*$7H_2O$, 40 mM $NaH_2PO4$*$H_2O$, 10 mM KCl, 1 mM $MgSO_4$*$7H_2O$), $OD_{600}$ recorded, and 400 μL cells added to 600 μL fresh Z-buffer. Cell suspension was lysed with addition of 40 μL chloroform, 20 μL 0.1% SDS, and vortexing followed by 10-minute incubation at 30˚C. β-galactosidase activity assay was begun with addition of 200 μL o-nitrophenyl-β-D-galactopyraniside (ONPG) and incubated at 30˚C for 20 minutes. If reactions become visibly yellow, they were halted immediately by addition of 500 μL $Na_2CO_3$ and the time recorded. All other reactions were halted in this manner after 20 minutes. Reactions were centrifuged to remove debris and 900 μL added to a cuvette and $A_{420}$ measured. LacZ reporter activity (MU) was calculated by: $MU = (A_{420} * 1,000) / (Time(mins) * 0.4 * OD_{600}$ of initial Z-buffer cell suspension).

### Statistical analyses and data handling

All statistical analysis and data handling were performed using R core statistical packages and the Dunn.test package. Shapiro–Wilk normality tests were performed to confirm non-normality of datasets. To compare group medians for more than 2 groups, a Kruskal–Wallis test with post hoc Dunn test and Benjamini–Hochberg correction was performed, with a $P \leq 0.025$ taken to indicate significance. For comparisons of 2 groups only, a two-sample Wilcox test was used. Spearman correlation was used to test significance of correlations between motility rate and rhamnose concentration, using the R core statistical function cor.test with the Spearman method specified.

For analysis of transcriptomic data, RNA sequencing output was run through Rockhopper for reference-based transcript assembly. Transcript count data were then analysed using DESeq2 package in R, for differential gene expression analysis, as well as generation of PCA and Volcano plots. A gene was taken as being differentially expressed if it differed by a Log2-Fold change of $\geq 1$ or $\leq -1$ between 2 samples (representing a doubling or halving of expression level, respectively) with an adjusted $P$ value of $\leq 0.001$.

To identify whether Rho-dependent terminators were present downstream of genes of interest, the RhoTermPredict algorithm [82] was run on the *P. fluorescens* SBW25 reference genome in Python.

## Supporting information

**S1 Fig. FleQ-homolog encoding gene *ntrC* is essential for rescued flagellar motility in *ntrB* mutant strain and depends on the presence of a kinase mutation.** Transcription factor gene *ntrC* was deleted and then reintroduced as a single-copy chromosomal insertion expressed from an L-rhamnose–inducible promoter system (*rhaSR-PrhaBAD*). The same complementation lacking the *ntrB*-T97P mutation as well as an empty expression system transposon were included as further controls. Photographs of motility after 1-day incubation in 0.25% agar LB plates supplemented with or without 0.15% L-rhamnose for induction of transcription factor expression.
(TIF)

**S2 Fig. PFLU1131-del15 mutation engineered into the Δ*fleQ* and Δ*fleQ*Δ*ntrC* ancestral lines provides motility in both conditions.** This indicates that rescue of motility by this mutation does not depend on Δ*ntrC* and is the sole mutation required to do so.
(TIF)

**S3 Fig. Impact of motility rescue mutations of flagellar and chemotaxis gene expression.** Log2Fold changes in gene expression relative to the Δ*fleQ* ancestor are shown for all genes associated with flagellar motility in *P. fluorescens* SBW25. Functional groups of genes are indicated by the coloured bars and labels to the left of the plot. Data underlying this figure can be found in S15 File.
(TIF)

**S4 Fig. Individual microbial growth curves that provide area under the growth curve (AUGC) values that are plotted in Fig 4C.** Six biological replicates for each of the 3 strains were assayed for change in $OD_{595}$ over 24 hours of incubation in shaking LB. AUGC values corresponding to each curve are displayed in the table below the graphs. Data underlying this figure can be found in S10 File.
(TIF)

**S5 Fig. Phenotypes of motility rescue second-step mutants. (A)** Race assay as measure of motility fitness. Distance moved over 24 hours in 0.25% agar LB plates measured relative to the Δ*fleQ* ancestor *ntrB*-T97P mutant. **(B)** Fitness in LB measured as area under the 24-hour growth curve (AUGC) relative to Δ*fleQ* ancestor *ntrB*-T97P. In both plots, mutations are coloured by functional category: red: PFLU1131/2; purple: PFLU1583/4; blue: metabolic; yellow: global regulatory; grey: other. For all boxplots: box represents first to third quartile range, middle line represents median value, whiskers range from quartiles to maxima and minima. Data underlying parts A and B figure can be found in S16 and S17 Files, respectively.
(TIF)

**S6 Fig. PFLU1583 Δ48–74 mutation depends to presence of PFLU1131 mutation and does not act through sigma factor RpoE. (A)** *rpoE* knockout does not revert PFLU1583 mutant to first-step motility phenotype. **(B)** Motility of PFLU1583 Δ48–74 with and without accompanying *PFLU1131*-del15 mutation.
(TIF)

**S7 Fig. Activity of LacZ expression reporter (Miller units (MU)) with increasing L-rhamnose concentration (%w/v) in both the Δ*fleQ* and Δ*fleQ*Δ*ntrC* ancestral genetic backgrounds with miniTn7[rhaSR-PrhaBAD-stRBS-LacZ].** Whiskers represent standard deviation above and below the mean value. Data underlying this figure can be found in S18 File.
(TIF)

**S1 File. Motile isolates were sampled and whole genome resequenced at each step.** This file provides the sequencing data. File can be access via OSF project https://osf.io/pcdhx/.
(XLSX)

**S2 File. Time to emergence and details of strains that failed to evolve motility within the 6-week cutoff.** File can be access via OSF project https://osf.io/pcdhx.
(XLSX)

**S3 File. Genome location information of mutations.** File can be access via OSF project https://osf.io/pcdhx/.
(XLSX)

**S4 File. Details of strains, plasmids, media supplements, and primers.** File can be access via OSF project https://osf.io/pcdhx/.
(XLSX)

**S5 File. Raw output from RNA-seq giving Log2Fold change in gene expression relative to ΔfleQ ancestor.** File can be access via OSF project https://osf.io/pcdhx/ and NCBI GEO GSE228016.
(XLSX)

**S6 File. Data underlying Figs 2C, 5B, 5C and 5D.**
(CSV)

**S7 File. Data underlying Fig 3.**
(CSV)

**S8 File. Data underlying Fig 4A.**
(CSV)

**S9 File. Data underlying Fig 4B.**
(CSV)

**S10 File. Data underlying Figs 4C and S4.**
(CSV)

**S11 File. Data underlying Fig 5E.**
(CSV)

**S12 File. Data underlying Fig 6B.**
(CSV)

**S13 File. Data underlying Fig 7A.**
(CSV)

**S14 File. Data underlying Fig 7B.**
(CSV)

**S15 File. Data underlying S3 Fig.**
(CSV)

**S16 File. Data underlying S5A Fig.**
(CSV)

**S17 File. Data underlying S5B Fig.**
(CSV)

**S18 File. Data underlying S7 Fig.**
(CSV)

## Acknowledgments

Bioinformatics analyses for the article were carried out using the Medical Research Council's (MRC) Cloud Infrastructure for Microbial Bioinformatics (CLIMB), Illumina Whole-Genome Sequencing conducted by MicrobesNG, Birmingham, UK (http://www.microbesng.com), which was supported by the BBSRC (grant number BB/L024209/1), and Illumina RNA sequencing with ribodepletion was conducted by Oxford Genomics Centre, Oxford, UK. We would like to thank Prof. Laurence Hurst for insightful discussion of earlier manuscript versions, Dr. Jonathan Nzakizwanayo and Dr. Iain MacArthur for providing us with *E.coli* strains SP50 pRK2073, S17 pTNS2, and ST18, Dr. Ellie Harrison for providing the R script that Fig 5A is based upon, and Dr. James Horton, Louise Flanagan, and Josie Elliott for helpful discussion during the course of the study.

## Author Contributions

**Conceptualization:** Matthew J. Shepherd, Tiffany B. Taylor.

**Data curation:** Matthew J. Shepherd, Aidan P. Pierce.

**Formal analysis:** Matthew J. Shepherd.

**Funding acquisition:** Tiffany B. Taylor.

**Investigation:** Matthew J. Shepherd.

**Methodology:** Matthew J. Shepherd.

**Project administration:** Tiffany B. Taylor.

**Supervision:** Tiffany B. Taylor.

**Writing – original draft:** Matthew J. Shepherd.

**Writing – review & editing:** Aidan P. Pierce, Tiffany B. Taylor.

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
