## [Editor Report · Decision Letter 0]

30 May 2023

Dear Dr Taylor, 

Thank you for submitting your manuscript entitled "Evolutionary innovation through transcription factor promiscuity in microbes is constrained by pre-existing gene regulatory network architecture" for consideration as a Research Article by PLOS Biology.

Your manuscript has now been evaluated by the PLOS Biology editorial staff, as well as by an academic editor with relevant expertise, and I'm writing to let you know that we would like to send your submission out for external peer review.

Once your full submission is complete, your paper will undergo a series of checks in preparation for peer review. After your manuscript has passed the checks it will be sent out for review. To provide the metadata for your submission, please Login to Editorial Manager (https://www.editorialmanager.com/pbiology) within two working days, i.e. by Jun 01 2023 11:59PM.

Kind regards,

Roli Roberts

Roland Roberts, PhD

Senior Editor

PLOS Biology

rroberts@plos.org

---

## [Decision Letter · Decision Letter 1]

8 Aug 2023

Dear Tiffany,

Thank you for your patience while your manuscript "Evolutionary innovation through transcription factor promiscuity in microbes is constrained by pre-existing gene regulatory network architecture" went through peer-review at PLOS Biology. Your manuscript has now been evaluated by the PLOS Biology editors, an Academic Editor with relevant expertise, and by four independent reviewers.

You'll see that reviewer #1 is very positive and only has a handful of minor textual issues. Reviewer #2 starts by explaining why (despite initial impressions) this work is important; however, s/he fails to be convinced by your evidence for the role played by promiscuity. Their objection seems to be that you repeatedly claim/assume “promiscuity” of TFs and kinases without actually demonstrating it; s/he makes some suggestions about how to get around this, semantically. Reviewer #3 is positive, but says that the paper would have been stronger if they’d provided some biochemical evidence for some of the proposed mechanisms; s/he also had problems with their use of “promiscuity,” has some presentational issues and wants you to tone down a claim. Reviewer #4 is fairly positive, but wants more discussion of the evolutionary significance of the work, and its limitations. S/he also questions the evidence for promiscuity, wants better treatment of the existing literature, and has some presentational requests.

I discussed the reviews with the Academic Editor, who said, "In a nutshell, I don't think the original storyline with an emphasis on transcription factor promiscuity holds. The authors have direct evidence on TF rewiring (i.e. new targets), but not on elevated levels of promiscuity (i.e. increased number of targets). It is doable with a range of HT techniques, but it is a substantial amount of work: in my eyes, at least 6 months, if properly done. A possible short-cut is changing the emphasis, as suggested [by reviewer #2], namely to change “promiscuity” (which implies a specific molecular mechanism) to “rewiring” (which is a more general term that can involve multiple mechanisms). In short, I would give light major revision and tone down the arguments on promiscuity throughout the ms." [comments lightly edited]

In light of the reviews, which you will find at the end of this email, we are pleased to offer you the opportunity to address the comments from the reviewers in a revision that we anticipate should not take you very long. We will then assess your revised manuscript and your response to the reviewers' comments with our Academic Editor aiming to avoid further rounds of peer-review, although might need to consult with the reviewers, depending on the nature of the revisions.

**IMPORTANT - SUBMITTING YOUR REVISION**

*Resubmission Checklist*

*Published Peer Review*

*PLOS Data Policy*

*Blot and Gel Data Policy*

Sincerely,

Roli

Roland Roberts, PhD

Senior Editor

PLOS Biology

rroberts@plos.org

REVIEWERS' COMMENTS:

Reviewer #1:

The manuscript of Shepherd et al., describes a study focussed on understanding gene regulatory network rewiring using an elegant experimental model. The authors describe a hierarchy of rewiring in transcription factors to rescue a loss of function mutation. The authors reveal key features for rescuing in GRN - high activation, high expression and low level affinities for target genes. These features reveal that effects can be targeted and global in terms of functional changes with pre-existing architecture of the GRN exerting influences over gain of function. This work has profound implications in terms of our understanding of evolvability of gene networks through TF innovation.

Overall the manuscript is beautifully written with a clear narrative.

There are a few minor points for the authors to clarify/modify, but otherwise the manuscript makes a significant contribution to understanding in this area and I congratulate the authors on a great piece of work.

Minor points

Line 83/84 - for non-specialists adding a few words about this strong selection for swimming and perhaps a reference could be useful.

Line 100 - would the sentence sound clearer if 'is the only reason' is replaced with 'Perhaps..'

Line 472-473 - I feel this is maybe underplayed a little, and perhaps the authors should make the distinction clearer that in two component systems rather than one-component systems (for want of a better phrase) for example metabolic responsive TFs - Ara, GntR, lac, Tet etc…this is explained a little in the paragraph starting line 495…interestingly, negatively autoregulatory TF maybe have smaller regulons? So perhaps loss of function mutations are more common? To diversify function in two component systems, the way to do this could actually to for heterologous RRs - see https://www.ncbi.nlm.nih.gov/pmc/articles/PMC4125116/

Reviewer #2:

This manuscript demonstrates how the loss of a specific regulatory gene can be compensated by secondary mutations that result in rewiring of gene regulatory networks and (partly) restore fitness.

The key idea of this study is not novel - several previous studies (several of which are duly cited) have already revealed that 1) evolution is often surprisingly repeatable and 2) that promiscuity greatly facilitates evolutionary adaptation. However, what sets this manuscript apart from much of the previous body of work is that it uses de novo adaptation (adaptive laboratory evolution) to investigate which transcription factors and regulatory modules in are most likely to drive cellular adaptation to a specific new condition. As is the case in may other studies, they find that evolutionary adaptation consistently and repeatedly follows a similar mutational trajectory. Importantly, they go on to study whether other adaptive pathways exist if the most preferred evolutionary route is closed off (by deleting the key transcription factor that is always mutated). They find that indeed, other pathways do exist, even if these are never chosen if the more preferred adaptive route exists, likely partly because the mutational target size or rate to obtain these alternative suppressor mutants is lower, and partly because they also suffer a fitness deficit compared to the mutants that follow the preferred mutational path.

As such, this study contributes to our general understanding of evolution by demonstrating a certain hierarchy or preference in the exact genomic modules (transcription factors) that drive evolutionary adaptation, and shows that the hierarchy is a result of both mutation supply and differences in fitness of the (intermediary) mutants. In addition, the study also confirms that promiscuity is a prime driver of evolutionary adaptation (although I think that this particular conclusion is not only not very novel, but also not very convincing because promiscuity is not directly demonstrated - see major comment below).

Major comments

My only major concern regarding this study is that the authors do not provide much experimental evidence for the molecular mechanism by which compensatory mutations work. Instead, they seem to assume that every compensatory mutation always increases "promiscuity"…

Throughout the text, the authors use the word promiscuous to denote promiscuous binding of transcription factors or kinase substrate specificity (established terminology) as well as a more general term that seems to refer to alternative regulation that may be due to rewired transcriptional networks or kinase specificity, but for which this has not been demonstrated directly. Instead, the authors seem to assume that the interactions become promiscuous, without providing evidence (eg in line 131, line 240, the title in 248, line 321, the whole paragraph starting at line 326, and line 402…). Unless I am mistaken, promiscuous binding of the transcription factors or kinases was not demonstrated, yet the authors seem to assume that this must be happening. I think it is not OK to call a protein or mutation "promiscuous" without directly showing that it changes the specificity of its activity or interaction. Instead, I would suggest to clearly separate the concept of a promiscuous interaction (between molecules) from the broader concept of changes in physiology and regulation that can be due to multiple mechanisms. May be better to call the latter "transcriptional rewiring" (an established term that the authors also use in the introduction)? 

The finding that increased expression of transcription factors may stimulate transcriptional rewiring is interesting and plausible. However, again, the authors do not directly show that this is indeed the case (by showing that the over-expressed transcription factor indeed shows more promiscuous binding).

Minor

The "Results" section starts a bit abruptly, without sufficient introduction. I think it is useful to inform the reader (again) about the outcome of previous experiments before asking the questions what the reason for this outcome (the repeated mutations in NtrC) could be…

Line 114: what is a "sensor histidine kinase"? Please explain in text.

It might be personal preference, but I am not a fan of naming the two adaptive evolutionary paths the "commonly rewired route" and the "unmasked rewiring pathway". What about "primary adaptive route" and "secondary adaptive route", or a name that alludes to suppression?

Line 237: typo - "severE"

Line 280 : act on RpoN, the partner sigma factor of PFLU1132, (add comma's)

Discussion section: It would be interesting to compare the results with that of a recent very similar study published in Molecular Biology and Evolution, where Helsen and coworkers perform a very similar series of experiments by evolving more than 200 parallel populations as they adapt to the loss of a specific gene (with multiple gene deletions being tested). Here too, the authors find reproducible adaptive mutational paths as well as primary and secondary mutations, with the latter ones restoring fitness. Moreover, they find that compensation for the loss of highly connected "hub" genes generates more in adaptive paths, as well as the fitness that the adapted mutants reach (reference: PMC7530610)

Reviewer #3:

This manuscript is a follow-up project from a study published in 2015. In that initial work, the transcription factor FleQ controlling flagellar synthesis in Pseudomonas fluorescence was deleted and the evolutionary rescue of the motility was studied. In all replicates, a homologous transcription factor, NtrC, was recruited to control the flagellar genes. 

Here, the same evolutionary experiment was performed, but this time in a strain that has FleQ and NtrC deleted. This time, another homologous transcription factor, namely PFLU1132, was recruited to control the flagellar genes. Mutations are identified through genome sequencing and gene expression levels are monitored by RNA sequencing. The authors try then to draw general conclusions about what factors favor the recruitment of a homologous transcription factor and why this happens faster/more frequently for NtrC than for PFLU1132. They conclude that for NtrC provides a better and easier reached solution. They also conclude that increasing activation and expression of the recruited transcription factors helps to increase the off-target activity.

This manuscript addresses an important question. It presents a nice system to study this question and the findings are interesting. The study could have been even stronger, if combined with some biochemical analysis to confirm putative mechanisms, such as phosphorylation levels or confirmation of putative binding sites or putative increased readthrough. 

Nevertheless, I support publication if my comments are addressed. 

My comments mainly aim to increase clarity and understandability, as well as tuning down some claims.

Major comments:

* The manuscript talks many times about "gaining promiscuity". For me, changing promiscuity is changing the binding affinity of the transcription factor to the DNA by either mutating the protein or the DNA binding site. However, if I understood correctly, neither of them happens here (in contrast to the initial study where the transcription factor NtrC acquires a mutation in the DNA binding site). Here, it simply increases off-target binding by increasing the expression levels of PFLU1132 and its activity by phosphorylation via PFLU1131. I think this should be re-formulated throughout the manuscript to make clearer.

* Even though it is likely that the mutation in PFLU1131 increases phosphorylation of PFLU1132 and therefore its activity, the manuscript does not present any direct proof, e.g. by western blot analysis for the phosphorylated protein. I am not saying the authors have absolutely to do this, but I think the claims have to be tuned down. 

* Lines 166-174 is a repetition of the previous section. I found this very confusing. Find a better way to make the connection between the two sections.

* Often, the background in which a mutation was tested is not clearly indicated. E.g. Fig. 4BC. I believe PFLU1131-del15 is in a ΔfleQΔntrC background, but it is not mentioned. The same for Figure 5CD

* In Fig. 2B it is not at all clear which picture corresponds to which strain tested. In your preprint, the pictures are labelled (but also unclear, as the background is missing, see comment above), but in this version, I don't see any labels at all?!? Maybe make schematic drawings to make this clearer and easier to understand?

Minor comments:

* Separating figures from and captions from the main text makes the life of reviewers very difficult. For next time (especially for a first round), please embed the figures.

* In the beginning, please explain in few sentences why there is a strong selection for motility (local nutrients become depleted). It is not very clearly explained. It only become clear to me when reading the initial study.

* In the introduction and/or discussion, discuss the importance of gene duplication. Those transcription factors are homologs, i.e. generated by gene duplication. However, I did not read gene duplication a single time in the manuscript. I think this should be at least shortly elaborated. 

* In the discussion, I would also appreciate a comment about alternative mechanisms that could lead to rewiring, e.g. by mutation of the cis-regulatory elements.

* Line 179 refers to Supporting Fig. S3, but this figure shows growth curves related to Fig. 4. It should probably say Fig. S4.

* For the first time use (line 124), write SNP out

* Line 89: "additional transcription factors" it is only one (PFLU1132), no?

* Line 118: quickly explain a two-component system. Not everybody will be familiar with it.

* After Supporting Figure S2 you cite Supporting Figure S4. Usually, they should named in order of appearance.

* Line 183 cites ref. 34, but I think it should be ref. 31

* Figure caption 3: "ΔfleQΔntrC first step" is a strange expression. I get what you mean, but it would be more precise to write something like "strain with mutations acquired in the first step, i.e. ΔfleQΔntrC PFLU331-del15"

* ΔfleQ ntrB-T97P appears the first time in Figure 3 and then Figure 4. It needs some explanation in the main text. 

* Fig. 4A. Can you do statistical tests comparing the two 1st steps together and another comparing the two 2nd steps together? That would seem more relevant

* Line 200. I think instead of "considered" measured or compared would be more appropriate

* Line 243: For the cells having a low pleiotropic fitness cost is in principle a good thing, so that per se does not make it a poorer solution. It is just an indication that the expression level is lower

* Fig. 6, maybe also indicate the putative positive feedback loop. It took me a while to understand RpoN binding site would mean that PFLU1132 might activate transcription there.

* Line 470: It is not only the topology of the GRN (e.g. negative vs positive feedback), but also the local sequence context that matters. For example, the increase in read-through by deleting a terminator is not really connected with the network topology (architecture), but the local sequence context.

* Supporting Fig. S3: Why do some growth curves start below 0? Did you also try other measures to analyse these curves, such as growth rates? Are your conclusions robust to the measure you choose?

* Supporting Fig. S6. I believe A and B are inversed (either in the caption or in the figure, but the two don't fit together).

Reviewer #4:

The authors present a study on gene regulatory network "rewiring" (i.e. a change in GRN connections) mediated by transcription factor promiscuity in the bacterium Pseudomonas fluorescens. They find that there is a hierarchy among TFs that are rewired to rescue lost function if the primary pathway is eliminated and identify three key properties (high activation, high expression, and low-level affinity for novel targets) that determine TF factor evolvability. While gene network evolution is an interesting, timely, and important topic, my concerns below must be addressed before I can support publications in PLoS Biology.

Major Concerns:

-I struggle to see the biological/evolutionary relevance of this study. I am not clear of why GRN re-wiring is relevant to the experimental model system considered. If there was a fitness disadvantage in terms of Pseudomonas fluorescens' ability to swim for all rewired networks, under which scenarios would these alternative networks evolve? Even if a rewiring mutation occurred in a small number of cells in population it would be selected against and disappear from a microbial population unless there was a fitness advantage or in a regime where genetic drift applies. Also, what is the motivation for investigating the evolutionary rescue of a non-RpoN dependent trait? Finally, the authors should discuss the limitations of their study; specifically that their findings are specific to the GRNs with transcription factors homologous to the target transcription factor.

-If high expression of the transcription factors (PFLU1132 or ntrC) lead to promiscuity, how do you know promiscuity is not facilitated due to the expression of these genes, rather than promiscuity being facilitated by the saturation of the cognate binding sites? Also, are you sure that the GRNs are being re-wired as opposed to independent/compensatory mutations arising to restore the slow and fast swimming phenotypes?

-The results of this study are not properly put into context of previous work. 1) There are several relevant publications from Mads Kaern and Gabor Balazsi's groups on how GRN motifs affect drug resistance evolution (see Charlebois et al., Phys. Rev. E, 2014; Gonzalez et al., Mol. Syst. Biol., 2015; Farquhar et al., Nat. Commun., 2019; Camellato et al., Eng. Biol., 2019) that should be incorporated into the manuscript; 2) The authors have previously published a study on PFLU1131 (https://doi.org/10.1093/molbev/msac132), which they should discuss when they state that PFLU1131 remains unstudied; and 3) MacLean et al., PNAS, 2004 should be cited when discussing pleiotropic fitness cost and central carbon metabolism.

Minor Concerns:

Why was there no PFLU1131 knockout created? Perhaps PFLU1132 could have worked with another component?

It is stated multiple times that the pathway of ntrBC is always utilized even in the presence of the other PFLU1131/2 pathway, yet in line 166, it is stated that the first step PFLU1131 mutation would lead to motility even if ntrC is present?

Introduction, first paragraph: The homologous transcription factor and its relation to the FleQ and ntrC is not clear. Mention specifically on how they are structurally related, and how the level of homology is determined.

Line 67: Provide a biological example of transcription factor promiscuity to motive the study, in addition to the general statement/references provided. 

-Section on "Motility evolution experiments": How exactly is motility determined and how is this differentiated from colony expansion due to cell division? The authors provide references to previous work, but it would be helpful if these details were explicitly included in the manuscript.

Figure 2: Caption needs to be improved to include more details. I am not clear what each panel B is supposed to show? What do black data points denote? Same comment for Fig. 5C,D.

-Figure 3 caption, second sentence: Should read "ΔfleQΔntrC and ΔfleQΔntrC del" and not "ΔfleQΔntrC and ΔfleQΔntrC"?

Style & Grammar

-Use quotations the first time you use words like "circuits", "wires", "crosstalk" etc. in a biological context.

-Line 363: "PFLU1131/2" not "PFLU113/2".

-Line 462: Define CF.

-Line 501: "enable innovation"?

-Discussion: Remove bolded text.

-To enhance clarity, I suggest that the paragraphs explaining the two step mutations should come before all the other paragraphs discussing first or second step mutations.

-Figure captions: Be consistent with formatting and clearly state the intended message of each figure in the figure title/first sentence of the caption.

---

## [Editor Report · Decision Letter 2]

14 Sep 2023

Dear Dr Taylor,

Thank you for your patience while we considered your revised manuscript "Evolutionary innovation through transcription factor rewiring in microbes is constrained by pre-existing gene regulatory network architecture" for publication as a Research Article at PLOS Biology. This revised version of your manuscript has been evaluated by the PLOS Biology editors and the Academic Editor.

Based on our Academic Editor's assessment of your revision, we are likely to accept this manuscript for publication, provided you satisfactorily address the following data and other policy-related requests.

IMPORTANT - please attend to the following:

a) We note that the Title seems somewhat self-evident, and we wonder whether you would be able to modify it to more explicitly include some ideas of which attributes of pre-existing GRN architecture are important (i.e. the three "key properties" mentioned in the Abstract)?

b) Please include the study species in the Abstract.

c) Please address my Data Policy requests below; specifically, we need you to supply the numerical values underlying Figs 2C, 3, 4ABC, 5BCDE, 6B, 7, S3, S4, S5, S7, either as a supplementary data file or as a permanent DOI’d deposition. I note that you already have an associated OSF deposition, so you could add the underlying data to those files (some of which may already contain data relating to the Figs) or provide extra supplementary files.

d) Please cite the location of the data clearly in all relevant main and supplementary Figure legends, e.g. “The data underlying this Figure can be found in S1 Data” or “The data underlying this Figure can be found in https://doi.org/XXXXX"

e) Please make any custom code available, either as a supplementary file or as part of your OSF deposition.

We expect to receive your revised manuscript within two weeks. 

*Published Peer Review History*

*Press*

Sincerely,

Roli Roberts

Roland Roberts, PhD

Senior Editor,

rroberts@plos.org,

PLOS Biology

DATA POLICY:

Regardless of the method selected, please ensure that you provide the individual numerical values that underlie the summary data displayed in the following figure panels as they are essential for readers to assess your analysis and to reproduce it: Figs 2C, 3, 4ABC, 5BCDE, 6B, 7, S3, S4, S5, S7. NOTE: the numerical data provided should include all replicates AND the way in which the plotted mean and errors were derived (it should not present only the mean/average values).

CODE POLICY

Per journal policy, as the code that you have generated is important to support the conclusions of your manuscript, we require that you make it available without restrictions upon publication. Please ensure that the code is sufficiently well documented and reusable, and that your Data Statement in the Editorial Manager submission system accurately describes where your code can be found.

SPECIES INDICATED IN THE ABSTRACT? 

- Please note that per journal policy, the model system/species studied should be clearly stated in the abstract of your manuscript. 

We require the original, uncropped and minimally adjusted images supporting all blot and gel results reported in an article's figures or Supporting Information files. We will require these files before a manuscript can be accepted so please prepare and upload them now. Please carefully read our guidelines for how to prepare and upload this data: https://journals.plos.org/plosbiology/s/figures#loc-blot-and-gel-reporting-requirements

DATA NOT SHOWN?

---

## [Editor Report · Decision Letter 3]

25 Sep 2023

Dear Dr Taylor,

Thank you for the submission of your revised Research Article "Evolutionary innovation through transcription factor rewiring in microbes is shaped by levels of transcription factor activity, expression and existing connectivity" for publication in PLOS Biology. On behalf of my colleagues and the Academic Editor, Csaba Pal, I'm pleased to say that we can in principle accept your manuscript for publication, provided you address any remaining formatting and reporting issues. These will be detailed in an email you should receive within 2-3 business days from our colleagues in the journal operations team; no action is required from you until then. Please note that we will not be able to formally accept your manuscript and schedule it for publication until you have completed any requested changes.

Sincerely, 

Roli Roberts

Senior Editor

PLOS Biology

rroberts@plos.org